# Biomimetic Approach for Enhanced Mechanical Properties and Stability of Self-Mineralized Calcium Phosphate Dibasic–Sodium Alginate–Gelatine Hydrogel as Bone Replacement and Structural Building Material

**Alberto T. Estevez ***(ID) **and Yomna K. Abdallah ***(ID)

IBAG, Institute for Biodigital Architecture and Genetics, School of Architecture,
Universitat Internacional de Catalunya, 08017 Barcelona, Spain
* Correspondence: estevez@uic.es (A.T.E.); yomnaabdallah@uic.es (Y.K.A.)

**Abstract:** Mineralized materials are gaining increased interest recently in a number of fields, especially in bone tissue engineering as bone replacement materials as well as in the architecture-built environment as structural building materials. Until the moment, there has not been a unified sustainable approach that addresses this multi-scale application objective by developing a self-mineralized material with minimum consumption of materials and processes. Thus, in the current study, a hydrogel developed from sodium alginate, gelatine, and calcium phosphate dibasic (CPDB) was optimized in terms of rheological properties and mineralization capacity through the formation of hydroxyapatite crystals. The hydrogel composition process adopted a three-stage, thermally induced chemical crosslinking to achieve a stable and enhanced hydrogel. The 6% CPDB-modified SA–gelatine hydrogel achieved the best rheological properties in terms of elasticity and hardness. Different concentrations of epigallocatechin gallate were tested as well as a rheological enhancer to optimize the hydrogel and to boost its anti-microbial properties. However, the results from the addition of EPGCG were not considered significant; thus, the 6% CPDB-modified SA–gelatine hydrogel was further tested for mineralization by incubation in various media, without and with cells, for 7 and 14 days, respectively, using scanning electron microscopy. The results revealed significantly enhanced mineralization of the hydrogel by forming hydroxyapatite platelets of the air-incubated hydrogel (without cells) in non-sterile conditions, exhibiting antimicrobial properties as well. Similarly, the air-incubated bioink with osteosarcoma SaOs-2 cells exhibited dense mineralized topology with hydroxyapatite crystals in the form of faceted spheres. Finally, the FBS-incubated hydrogel and FBS-incubated bioink, incubated for 7 and 14 days, respectively, exhibited less densely mineralized topology and less distribution of the hydroxyapatite crystals. The degradation rate of the hydrogel and bioink incubated in FBS after 14 days was determined by the increase in dimensions of the 3D-printed samples, which was between 5 to 20%, with increase in the bioink samples dimensions in comparison to their dimensions post cross-linking. Meanwhile, after 14 days, the hydrogel and bioink samples incubated in air exhibited shrinkage: a 2% decrease in the dimensions of the 3D-printed samples in comparison to their dimensions post cross-linking. The results prove the capacity of the developed hydrogel in achieving mineralized material with anti-microbial properties and a slow-to-moderate degradation rate for application in bone tissue engineering as well as in the built environment as a structural material using a sustainable approach.

**Keywords:** bone replacement materials; mineralized materials; biomineralization; calcium phosphate reinforced hydrogels; multi-scale applications; multi-disciplinary; biomimetic structural materials; bone hierarchical structure motifs

## 1. Introduction

Biomineralized materials are gaining increased interest in various fields, extending from regenerative medicine applications in bone tissue engineering to optimized structures, with large industrial applications such as architectural building blocks (bricks) [1–4] thanks to their strength and enhanced mechanical properties. By definition, biomineralization is the process by which living organisms produce minerals to compose hardened mineralized tissues [5]. For example, calcium phosphates and carbonates are involved in forming structures of shells and the bone in vertebrates. These biominerals, namely phosphate and carbonate salts of calcium, which are usually crystalline, often exist in combination with organic polymers like collagen and chitin to perform as structural support for bones and shells [6]. These bio composite materials are organized in multi-scale hierarchical structures from the nanometer to the macroscopic level, providing intricate architectural aesthetics as well as multifunctional applications. Including materials and tissue engineering applications in regenerative medicine as well as structural efficiency is feasible thanks to their material–structure function and multi-scale complexity. This has triggered research interest in analyzing and mimicking the biological mechanisms of biomineralization [7,8].

Typically, a biomineralization process signifies mineralization induced under biological control of mineral morphology, growth, composition, and site as ruled by the cellular processes of a specific organism, such as the mineralization of collagen granting bone, cartilage, and teeth with the adequate compressive strength and mechanical performance [9]. However, the definition of biomineralization induction in bone replacement material engineering can extend to involve the biomimetic approach of synthesizing or inducing the mineralization reaction to form these biogenic minerals of calcium phosphates and carbonates mineralized crystals [10,11]. Thus, in the current study, the authors focused on developing a biometric approach of biomineralization induction of calcium-phosphate-based hydrogels without the use of cells vs. with the use of embedded cells.

The main marker of the successful biomineralization induction is the formation of mineralized hydroxyapatite crystals. Hydroxyapatite is the most common biogenic form of calcium phosphate, with the chemical formula ($Ca_{10}(PO_4)_6(OH)_2$), and it is a naturally occurring form of apatite and is also the primary constituent of bone: 65 to 70% of bone mass is HA [12]. The biomimetic mineralization induction reaction tries to mimic the natural mechanism of remineralization, where the mineral ions are introduced to reinstate the structure of the hydroxyapatite crystals [13].

To achieve this goal, the current study introduces an optimized and sustainable hydrogel composition based on calcium phosphate by proposing a three-stage, thermally catalyzed cross-linking of the developed 3D-printed hydrogel and detecting its mineralization by the formation of mineralized hydroxyapatite crystals across multiple time intervals and in different incubation conditions. Furthermore, the mechanical properties of the developed hydrogel were tested to draw a picture about its rigidity vs. elasticity. In this paper, these tests are described, along with the same hydrogel composition with embedded osteosarcoma cells, to compare the mineralization efficiency and duration between the passively mineralized hydrogel (without cells) and the actively mineralized bioink (with cells).

To understand the criteria of selection of the hydrogel constituents, the following introduction exhibits the most popular adopted hydrogels in bone tissue engineering, which are based on alginate–gelatine composition, and their compatibility with calcium phosphate for inducing mineralization.

*Alginate–Gelatine–Calcium Phosphate Trio-Hydrogels for Bone Tissue Regeneration*

Bone tissue engineering is a promising discipline to provide solutions for bone defects [14]. Bone tissue regeneration is controlled by intracellular functions as well as the biophysical traits of the extracellular matrix (ECM) and is mainly affected by these mechanical indications in the surrounding microenvironment. Natural bone regeneration signifies a stiffening physiological process combining viable cells, extracellular matrix,

specific cytokines, and growth factors that formulate sufficient mechanical and vascular environments. This is usually tackled in bone tissue bioengineering research by employing bioinks encapsulating specific bioactive cells and molecules for application as grafts in the damaged area. During the bone regeneration, the ECM is dynamically remodeled with structural and compositional changes, resulting in dynamic stiffness alteration. This alteration has been proven to induce different biological behaviors of cells [15,16]. For example, it was reported that a matrix stiffness of 10–30 kPa induces MSCs differentiation into osteoblasts in two-dimensional (2D) microenvironment. Also, a matrix stiffness of 134 kPa endorses the excretion of alkaline phosphatase (ALP), osteopontin (OPN), and osteocalcin (OC) from osteoblasts when cells are cultured on 2D substrates [17]. Therefore, it is more relevant to design biomaterials that are dynamically changing in terms of stiffness to simulate the mechanical microenvironment in vivo while analyzing the effect of ECM dynamic hardening on bone regeneration to promote adopting such biomaterials of special biophysical properties in tissue regeneration in vivo [14]. This necessitates that bone substitute materials (BSM) possess an osteo-inductive 3D structure to encompass osteogenic cells and osteo-inductive factors, to offer sufficient mechanical properties, and to promote vascularization [18].

Hydrogels as high-water-content bio composite materials are vastly applied in tissue engineering scaffolds thanks to their efficient performance in diffusing nutrients and oxygen into their structure, simulating the biological tissues. This is enabled by their soft nature, elasticity, and permeable structure [18–20]. In addition to their viscoelastic traits and injectability, these hydrogels are compatible with additive manufacturing technologies (AMT) and 3D bioprinting. These hydrogels incorporating living cells form the construction blocks of three-dimensional scaffolds in bioprinting, which are called bioinks. Furthermore, hydrogels are ideal for developing scaffolds that mimic the ECM conditions thanks to the controllability of their mechanical properties by the dynamic tuning of their cross-linking degree [21]. It was reported that some cross-linking strategies dynamically changed hydrogels' stiffness under external stimuli. For example, temperature was applied to change hydrogel stiffness through controlling the viscoelasticity of hydrogels [14].

Commonly, interpenetrating network hydrogels are broadly employed to construct biomaterials with dynamic matrix stiffness for 2D and 3D cell culture, especially alginate-based hydrogels, which are the most tested in bone tissue engineering (BTE) applications and 3D bioprinting, thanks to their gelling property, low toxicity, high obtainability, and low cost. They provide an appropriate niche for cell embedding, which, in addition to their intrinsic ionic cross-linking, qualifies them as injectable hydrogels and makes them more favorable than other solid-scaffolds materials. Alginate polysaccharide is safe, according to the FDA (Food and Drug Administration) [22], for application in humans [23]. It is also compatible with 3D bioprinting techniques, which widens the opportunity to expand its applications in bone regeneration [18].

Alginate is chemically composed from a linear polysaccharide of homopolymer units of 1,4-linked (-D-mannuronic acid) (M) and (-L-guluronic acid) (G) [24]. M block segments possess a lined and flexible configuration, whereas the $(1\rightarrow4)$ linkages to guluronic acid introduce a steric disruption around the carboxyl groups. Therefore, the G block parts deliver folded and stiff structural configuration, giving the molecular chains their stiffness. High-M-content alginates promote the production of cytokine more than the high-G-content alginates [25]. Alginates originating from different sources possess different M and G contents and varied lengths of each block, affecting the final properties of the material [24]. Aqueous alginate solutions are non-Newtonian fluids [26] and become more viscose under lower pH values, obtaining maximum values at pH of 3.0–3.5. This response of alginates to different values of pH is attributed to the carboxyl groups on the alginate backbone. This pH-sensitive behavior is proven by the higher swelling ratios observed at higher pH values [18].

Alginate-based materials also enjoy the capacity of in situ gelation [27], solubility in water [28], cytocompatibility [29], mucoadhesive properties [30], release of active agents [31], and acting as a protective shield for cell and particle release systems [32]. However, their low degradation caused by the use of high-molecular-weight alginates to obtain mechanical properties similar to those of hard tissues remains a challenge for their use in vivo as bone replacement materials [33]. For example, higher-molecular-weight alginates produce stiffer gels, while medium- and low-molecular-weight alginates produce more degradable hydrogels with higher rates of cell proliferation. To moderate the molecular weights of alginate-based hydrogels, there are various methods, such as enzymatic preparation [34], ultrasonic irradiation [35], ultraviolet photolysis [36], oxidative-reductive depolymerization, and thermal degradation [18,37]. Through the past decade, multiple studies have attempted to achieve equilibrium between the mechanical traits and the degradation kinetics (and the resulting potential toxins from the degradation process) of alginate hydrogels by varying cross-linking methods, alginate molecular weight, chemical structure and processing technologies by tuning alginate hydrogels' chemical composition and structure to tailor specific properties for their applications in tissue engineering to improve degradability, mechanical properties, cell adhesion, and special features such as printability. Usually, this tuning occurs either through the carboxyl or the hydroxyl groups. For example, phosphorylated alginates were developed through the hydroxyl modification to induce hydroxyapatite nucleation for bone tissue regeneration [38].

Recently, oxidized alginates (OAs) are becoming widespread thanks to their highly reactive groups and faster degradation rates than the non-modified alginates. Alginate oxidation is usually considered a simple process with easy purification and non-toxic effects. The oxidation is conducted by reacting alginate with sodium periodate to produce two aldehyde groups in each oxidized unit by breaking the carbon–carbon bonds in the alcohol groups; consequently, a lower molecular weight and the oxidized uronate residues make OA susceptible to alkali-catalyzed elimination [39]. However, OA is less efficient in mechanical behavior than the unmodified alginate hydrogels. For example, Ref. [40] conducted the oxidation of sodium alginate at 4.9% to evaluate its degradability and in vivo reactions, where they reported the decrease in the molecular weight of the OA from 390 kDa to 255 kDa and a decrease in the compressive modulus from $150 \pm 13$ kPa to $754 \pm 21$ kPa in comparison to the non-oxidized gels. This alteration in the compressive strength during degradability implies an increase in the degradation rate, starting at the 6th day of incubation. However, the OA hydrogels maintained their gelling capacity in the presence of calcium ions via ionic bonding. Nevertheless, the ionic cross-linking of the reported OA hydrogels through the interaction between the polymer chains and the divalent ions decreased due to the reduced GG blocks in OA chains. Gomez, C.G. etc., 2007 [41] reported that alginate oxidation is selective towards the guluronate units, decreasing the ionically gelation at oxidation values higher than 10%. In addition, the oxidation degree is limited by the formation of hemiacetal groups and their interference in the reaction [42]. Due to the several aldehyde groups present in the backbone of OA hydrogels, they are highly active in covalent bonding, which alters the final polymer properties. The oxidation degree of covalently bonded hydrogels controls the network cross-linking mark, density, mechanical characteristics, degradation rate, and swelling behavior [39].

The ionic cross-linking of alginate with divalent cations is realized by the cooperative interaction with G monomers blocks to form ionic bridges, while M monomers blocks formulate weak junctions with divalent cations, unlike the tight junctions between G monomers blocks and divalent cations. The cross-linking mechanism occurs through the organization of divalent ions with four-carboxyl groups to form an egg-box like arrangement [43]. Thus, ionic gelation of alginate hydrogels has two options: internal or external gelation. Internal gelation develops in situ gelling hydrogels and is suitable for injectable alginate applications. In internal gelation, divalent cationic salts of low solubility at neutral pH are used, followed by a subsequent acidification of the medium to release the cations

to further control the gelation kinetics and result in homogeneous gels. For example, Ref. [44] evaluated the composition of alginate microspheres with calcium chloride and zinc sulphate salts for drug release applications, where Zn cations interacted with alginate at different sites with respect to Ca cations and were less selective but allowed a higher degree of cross-linking. This proves that the higher specificity of calcium ions in alginate cross-linking produces a harder and more rigid polymeric network. On the other hand, external gelation ionic cross-linking is employed to synthesize materials for bone tissue engineering applications within a simple, low-cost process thanks to the fast interaction between divalent ions and alginate. However, it results in non-homogeneous hydrogels with low mechanical properties due to the non-cross-linked alginate zones within the cross-linked alginate layers. Currently, several researches are attempting to alleviate the disadvantages of this gelling mechanism. Figure 1 represents the chemical structures of alginate and the gelation of alginate in the presence of calcium ions as a cross-linking process, showing the famous egg-box gelation model.

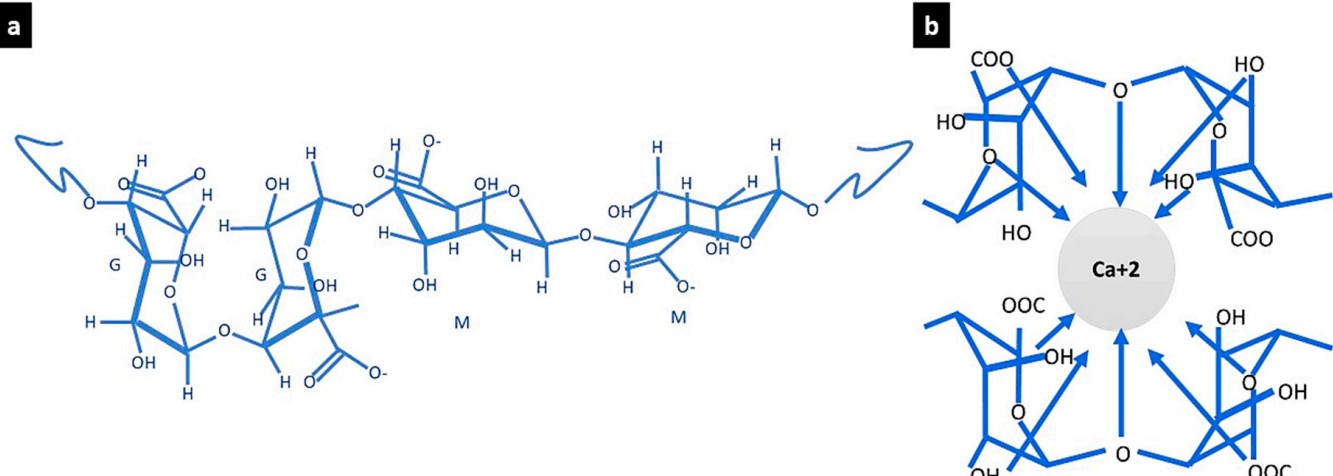

**Figure 1.** The chemical structure of alginate and its gelation by ionic cross-linking: (**a**) the chemical structure of alginate following [45] and (**b**) the egg-box model of the alginate ionic cross-linking by calcium ions following [18].

Sodium alginate (SA) is widely used to prepare biomaterials with dynamic mechanical properties. SA usually reacts differently to different concentrations of $CaCl_2$ or calcium ion chelating agents where the SA-based hydrogels are either stiffened or softened. Consequently, some studies recommended the continuous supplementation of calcium ions in SA-based hydrogels to sustain the cross-link degree, which could dynamically improve the stiffness of the obtained gel [14].

The covalent cross-linking of hydrogels results in strong and non-reversible chemical bonds that control the mechanical properties of alginate-based hydrogels [46]. On the other hand, these covalently bonded hydrogels usually suffer from lower injectability for in situ scaffolds, which hinders their application. Thus, another line of research to solve the in-situ gelation is by the generation of shear thinning viscoelastic hydrogels, including solutions of certain proteins, colloidal systems, peptides, and polymer mixtures with a self-assembly behavior that results in gelation of the material when it is not under shear stress [47]. Based on this property, the highly viscous polymer solution flows when a shear force is applied through injecting and forms a hydrogel when it stops. Alginates are an excellent base material for the generation of this type of materials due to their polyelectrolytic properties.

Alginates suffer from low cellular attachment due to their lack of any specific signal receptors for binding proteins, which makes them susceptible to unwanted interactions, immunological response, and proteolytic degradation [48]. To avoid this deficiency in alginates, researchers have combined oxidized alginates with proteins like collagen, and

gelatine [49,50]. Collagen-derived gelatines enjoy unique gelling traits due to the physical cross-linking of the triple-helix structure of the native collagen. The gelatines retain informational signals, and they are highly degradable in vivo while being controllable in terms of tuning their physicochemical properties. These advantages added to their high cellular adhesion, pliability, availability, and low cost have promoted the use of gelatines for cross-linking with OA, exhibiting high cytocompatibility with a wide range of cells strains [51,52]. Still, the end characteristics of the oxidized alginate-based hydrogels depend largely on the oxidation degree of the alginate since it was reported that a greater oxidation degree induces faster gelling, higher cross-linking degree, and lower swelling [18].

However, raw alginate hydrogels are limitedly applied in tissue engineering due to their lack of specific signal receptors for binding proteins, their poor cellular adhesion, and the difficult control of their degradation kinetics. However, they can be tuned chemically by modifying the alginate hydrogel composition and/or the covalent bonding design. On the other hand, human bone-like biomimetic mechanical properties have not been reached with alginate-based composites, nor have they been sufficiently explored in the injectable hydrogels field. Thus, more research is required to overcome these disadvantages.

Most of the literature focuses on studying how the chemical composition and architecture influence cellular phenotypical expression, differentiation, proliferation, migration, and ECM composition during in vitro and in vivo assays [53]. However, the used cell line and its interaction with the material should also be considered. For example, Ref. [54] developed insitu gelling alginate hydrogels encapsulating MC3T3 cells under controlled gelation kinetics. In terms of materials science, a single material can hardly provide all of the required properties for biomimetic bone replacement materials. Thus, the designing of composite materials emerged to combine the merits of different types of materials [55,56]. One biomimetic approach is the combination of organic and inorganic components for manufacturing materials similar to bone tissue. Bio glasses (BG) and calcium phosphates are the most used inorganic compound for the reinforcement of hydrogels.

Bioactive glasses, which are silica-based materials, have a high surface reactivity, which enhances promoting the nucleation and subsequent formation of calcium phosphate crystals, especially bonelike apatite crystals, on their surface when incubated in simulated body fluids [57,58]. Calcium phosphate ceramics are also highly employed in bone tissue engineering thanks to their bio conductive nature and osteoconductive and biodegradation properties depending on their chemical and physical parameters. The bioactivity of calcium phosphates can be explained by their dissolution releasing calcium and phosphorus ions into the microenvironment of the cells, promoting the formation of new bone, which is also supported by the rough surface of the particles [18,59].

For example, Ref. [60] prepared composite spongy scaffold from alginate and hydroxyapatite through a phase separation method, significantly improving the mechanical strength (compressive and elastic modulus) and enhancing the osteosarcoma cell adhesion and proliferation on the scaffolds. These scaffolds had an average pore size of 150 μm and over 82% porosity. Another study by [61] fabricated oxidized alginate–gelatine–biphasic calcium phosphate hydrogel scaffolds loaded with different contents of spherical HAp particles, where the scaffolds without HAp had higher porosity and interconnected pore structure than materials loaded with HAp (pore size average ≈100–250 μm). The scaffolds with higher granules resulted in a higher compressive strength (dry/compressive modulus $2.45 \pm 0.19$ MPa; wet/compressive modulus $51 \pm 0.39$ MPa). HAp-enhanced hydrogels exhibited the high cell viability, cell attachment, proliferation, and differentiation behavior of rat bone marrow-derived stem cells (BMSC). Perez, R.A. etc., 2012 [62] developed a core shell design of fibrous scaffolds made of alginate with $\alpha$-tricalcium phosphate for in situ cytochrome C protein loading and controlled delivery, which was produced by the injection of the alginate-$\alpha$-tricalcium phosphate and posterior cross-linking with $CaCl_2$. The hydrogel exhibited an initial drug delivery burst at lower cross-linking time and $CaCl_2$ concentration, and the scaffolds showed more elastic than viscous behavior. Scaffolds with higher amounts of $\alpha$-TCP yielded higher mechanical properties, with the storage modulus

increasing from ~80 kPa without $\alpha$-TCP to 800 kPa with 75 wt % $\alpha$-TCP, signifying that the $\alpha$-TCP phase promotes solidifying in the alginate matrix.

However, calcium phosphate bone cements still have some limitations that halt their clinical applications. For example, their mechanical strength is usually insufficient to provide adequate mechanical support to the defect site, having poor washout resistance and handling properties [63,64] reported that the addition of alginate increases the stiffening rate of calcium phosphate cements, resulting in quicker setting times. The compressive strength increased with incubation time, obtaining the highest value at 3 days and maintaining it up to 7 days of incubation. Particularly, compressive strength significantly increased from ~15 MPa to 60–70 MPa after 3–7 days of incubation at higher contents of alginate. Also, significant rat bone marrow-derived stromal cell adhesion, proliferation, and increased bone-associated gene markers such as collagen type-1, osteopontin, and bone sialoprotein were observed.

Thus, it can be concluded that developing sodium alginate hydrogels enhanced with gelatine and calcium phosphate can provide an integral solution balancing the required mechanical strength and cytocompatibility in bone tissue engineering, which is the main approach in the current study.

Furthermore, tissue engineering requires the development of scaffolds with a complex, customized geometry in combination with a precise control over the internal architecture and maintaining cytocompatibility. Bioprinting offers an integrated solution to the direct manufacturing of constructs with the possibility of incorporating cells and substrate materials including different kinds of additive manufacturing technologies (AMT) [65]. However, the type of AMT depends to a great extent on the processability of the bioink. Ribeiro, A.; etc., 2017 [66] defined printability as the possibility to extrude a hydrogel and dispense it in a pattern with a satisfactory degree of shape fidelity and also identify the accuracy of the printed structure in matching the original design. Printability is ruled by the rheological properties of materials and must be adjusted to the specific fabrication process to generate constructs with high shape fidelity [67]. Injectable alginate bioinks are one of the most used and successful materials for bioprinting due to their shear thinning character, rapid cross-linking ability, and feasibility of printing viable cells. They are particularly effective for bioprinting with nozzle systems due to the ability to protect the encapsulated cells through the process [27]. However, tuning alginate bioinks to meet specific requirements of printability, shape fidelity, and viability of encapsulated cells still requires further development to be used in the fabrication of effective 3D-printed constructs.

For instance, alginate bioinks must have sufficient viscoelasticity to achieve injectability during the printing process (rheological properties) and good shape fidelity to maintain the overall shape of the fabricated scaffold after printing [68]. Since the viscosity of alginate bioinks depend on the alginate concentration, the alginate molecular weight, and the cell density loading, printability can be promoted by controlling these parameters [27,69] Freeman and Kelly, 2017). A first approach is the control of alginate concentration. However, it can negatively influence the long-term biological performance at high concentrations. Park, J.; etc., 2018 [70] studied the influence of alginate concentration M/G 1.6 with high and low molecular weight (high $3.5 \times 10^5$ g/mol; low $1.43 \times 10^5$ g/mol) on printing fidelity and cell viability of a nozzle-based 3D printing system. Greater printing capacity, stability, and fidelity were obtained for hydrogels of high alginate of 3 wt% alginate and a 1:2 ratio (Low to High alginate). The metabolic activity and proliferation of the cells increased for lower molecular weights. Thus, high molecular weights produce dense networks and hinder mass transfer between the medium and the scaffolds, reducing the viability and proliferation of encapsulated cells.

Izadifar, M.; etc., 2017 [71] analyzed alginate bioinks' viscosity as function of temperature and polymer viscosity for low-concentrated alginate formulations. It was reported that bioinks prepared with 2% concentration of "medium-viscosity alginate" exhibited higher viscosity than formulations having a higher concentration of "low-viscosity alginate" in a range of temperature from 10 °C to 25 °C. Hybrid constructs reported cell viability >80%

for embryonic chick cartilage cells at 14 days. Hence, using alginate bioinks with high concentrations (i.e., 3.5–10 wt/v%) improves resolution printability and the material structural stability but limits cytocompatibility [70,71]

Another study by [72] developed alginate–gelatine blends by varying the individual constituent concentrations for study of printability, print accuracy, compressive behavior, and viability of encapsulated mesenchymal stem cells in bio-printed constructs. Higher concentrations of both alginate and gelatine resulted in printable bioinks with an optimal cross-linking time of 15 min in calcium chloride to improve stability per layer. In addition, the blends with 7% alginate–8% gelatine yielded high printability, mechanical strength, stiffness, and cell viability after printing. However, the compressive behavior of the hydrogel decreased rapidly over time and especially at 37 °C.

Added to the challenge of tuning rheological and mechanical properties of the hydrogel in balance with injectability, cell viability, and degradation, mineralization induction remains the biggest challenge in bone tissue engineering when developing bone replacement materials. Organic matrices offer this possibility based on tunable organic–inorganic interactions [73,74]. Therefore, further studies are required to optimize the osteo-inductive behavior of BRM. The conjunction of growth factors, bone cells, or bioactive materials where hydroxyapatite is used to promote the attachment of cultured osteoblasts in multiple polymers and improve their mechanical properties is a promising approach [75]. However, the alginate–hydroxyapatite system suffers from some limitations, including the difficulties of hydroxyapatite dispersion, the viscosity of the alginate, and the negative charge density of both that partially prevent their interaction. This hinders their processing together, especially for injectable materials. Alternatively, studies have reported the use of surface functionalization of hydroxyapatite with alginate or oxidized alginate since hydroxyapatite can be used as an effective sorbent and carrier for polymers thanks to its ion exchange capacity, adsorption capacity, and acidic properties [76]. These properties together with the alginate's ability to encapsulate cells and growth factors could encourage osteo-induction from simultaneous fronts. In addition, this overcomes the alginate hydrogels' limitation in reaching the structural properties required to function as mechanical support in bone tissue engineering, where the reported alginate–cement compositions achieved higher mechanical properties than alginate hydrogels (compressive strength 25–70 MPa; Young's modulus 90–130 MPa), although still far from the average mechanical properties of human bone (cortical bone: compression strength $200 \pm 36$ MPa; elastic modulus on compression $23 \pm 4.8$ GPa; cancellous bone-compression strength 1.5–38 MPa; elastic modulus on compression 10–1570 MPa) [77].

Thus, this alginate–gelatine cement still requires further experimentation to optimize the rheological and mechanical properties of the hydrogel, especially in the current study, where the indicated application is not only directed towards developing bone replacement materials but also developing self-mineralized material with a slow and low degradation rate that can be as well employed as a structural material in architectural and design applications. Since bioprinting can facilitate the tailoring of the properties of oxidized alginate-based hydrogels to the specific rheological properties of bone regeneration thanks to the ability to designing structures layer by layer and the ability to vary their processing conditions with a finer control over the principal alginate manufacturing conditions (addition of components, cross-linking agent concentration, and temperature), it was used in the current study to test and 3D print the developed hydrogels.

## 2. Materials and Methods

### 2.1. Rheological Properties of Alginate–Gelatine Hydrogel Enhanced with Different Concentrations of Calcium Phosphate

A hydrogel composed from 8% gelatine from bovine skin, type B in powder, and of pH 7 at 100 g/L at 60 °C was purchased from (G9391-Merck Life Science S.L. St. Louis, MO, United States), and + 4% sodium alginate E401 $(C_6 H_7 NaO_6)_n$ with molecular weight of 10,000–600,000 Daltons, purchased from (Quimics Dalmau SL, Barcelona, Spain), was

prepared in 60 °C distilled water stirred for 20 min before adding different concentrations of 4, 4.5, 5, 5.5, 6, 6.5, 7, and 7.5 g of calcium phosphate dibasic $CaHPO_4$ (CPDB) (C7263-Sigma Aldrich. St. Louis, MO, United States) and stirring the mixture for 4 h at 60 °C until reaching a homogenous gel, respectively. Each gel was tested by the syringe test before testing it for 3D printing using a Felix Bio dual-head extruder bio printer with a 5 mL syringe with a 0.5 mm nozzle. The objective was to print a $3 \times 3 \times 0.5$ cm$^3$ cuboid form with a square grid infill pattern with cell size of $0.5 \times 0.5$ mm$^2$. The 3D-printing process settings were 60 °C printing head temperature, 4 °C bed temperature, flow 105%, speed 100%, layer height 0.5 mm, and printing pressure 12 PSI. The 3D printing was conducted in sterile conditions. Post printing, each print was cross-linked with a 5 mL prepared solution of 5 mg calcium chloride $CaCl_2$ (449709-Sigma Aldrich. St. Louis, MO, United States) dissolved in 5 mL sodium carbonate concentrate $Na_2CO_3$ (72784-Sigma Aldrich. St. Louis, MO, United States). Three different hydrogels of the three different concentrations of 5.5, 6, and 6.5 g/100 mL calcium phosphate were selected to undergo the rheological tests and chosen based on comparison of their printability and post-printing shape retention and fidelity with the other gels containing the different concentrations of calcium phosphate. Three different thicknesses of 1, 3, and 6 mm of the 3D-printed geometry per each concentration were tested. The printed samples were left to dry in ambient temperature for a week before conducting the rheological properties tests. Each sample contained 10 specimens tested under the DISCOVERY HR-2 Rheometer manufactured by TA Instruments at the Biomaterials, Biomechanics, and Tissue Engineering division (BBT), Dept. of Material Science and Engineering (CEM), Universitat Polytechnic de Catalunya. The parameters $G'$ and $G''$ were estimated per each sample.

The geometry employed in the analysis consisted of a rugose plate with a diameter of 20 mm. The geometry was adjusted to specific gap heights for each sample height, specifically 3000 µm for samples with a height of 6 mm, 500 µm for samples with a height of 1 mm, and 700 µm for samples with a height of 3 mm. The testing sequence was then applied to each sample, employing a frequency of 1 Hz and a displacement ranging from $1 \times 10^{-5}$ rad to 10 rad at room temperature 25 °C. The values of $G'$ and $G''$ were recorded for each individual sample.

After that, the sample height of 1 mm was standardized for testing the rheological properties of the same three selected concentrations of 5.5, 6, and 6.5 g calcium phosphate. Each sample contained 10 specimens, and the test geometry employed in the analysis consisted of a rugose plate with a diameter of 20 mm. The test geometry was adjusted to 500 µm for all the samples, which had a height of 1 mm. The testing sequence was then applied to each sample, employing a frequency of 1 Hz and a displacement ranging from $10^{-5}$ rad to 10 rad, at room temperature 25 °C. The values of $G'$ and $G''$ were recorded for each individual sample.

*2.2. Rheological Properties of Alginate–Gelatine–Calcium Phosphate Hydrogel Enhanced with Different Concentrations of Epigallocatechin Gallate*

Three different concentrations of epigallocatechin gallate $C_{22}H_{18}O_{11}$ (E4143-Sigma Aldrich. St. Louis, MO, United States) of 4, 6, and 8 µm were added to the 6 g calcium phosphate-modified SA–gelatine hydrogel, respectively, to test the effect of EGCG on the printability, shape retention, and rheological properties of the developed hydrogel. The 3D-printing process employed identical settings as the previous test, and the tested geometry and the incubation conditions post printing were the same. Each sample corresponding to each concentration contained 10 specimens. The specimens were analyzed using the DISCOVERY HR-2 rheometer manufactured by TA Instruments. The parameters $G'$ and $G''$ of the three materials were examined. The test geometry employed in the analysis consisted of a rugose plate with a diameter of 20 mm and 1 mm height. The test geometry was adjusted to a force of 0.2 N for all the samples to determine the gap. The testing sequence was then applied to each sample, employing a frequency of 1 Hz and a displacement ranging from $10^{-5}$ rad to 10 rad. The values of $G'$ and $G''$ were recorded for each individual sample.

### 2.3. Mineralization Test of the CPDB-Modified SA–Gelatine without Cells, Incubated in Fetal Bovine Serum and Air, Respectively, for 7 and 14 Days, Respectively

The developed hydrogel with optimum concentrations of rheological enhancers was then tested for mineralization without the embedding of living cells (osteosarcoma SaOs-2 cells) to examine the hydrogel capacity of mineralization, being incubated in 10 mL of fetal bovine serum FBS each 3D bio printed sample following the same dimensions as described above and the same 3D printing settings (FBS was kindly provided from the GRC—Research Group in Regulation of Lipid Metabolism in Obesity and Diabetes at the International University of Catalunya UIC) or in the open air in non-sterile conditions. The hydrogel samples incubated in air included three specimens, which were incubated in the open air in ambient room temperature 25 °C for 7 and 14 days, respectively, while the incubated FBS samples included three specimens incubated in 25 °C for 7 and 14 days, respectively, with the FBS solution changed every 48 h. The samples and corresponding specimens were 3D printed in duplicates per each incubation condition (FBS and Air). The samples were tested using scanning electron microscopy (SEM) to detect the formation of hydroxyapatite crystals across these different time intervals. The sample preparation procedures for the SEM included preservation in glutaraldehyde and storing at 4 °C. Later, they were mounted on steel stubs with a bio adhesive layer and a glass cover and scanned under low vacuum for the SEM scanning.

### 2.4. Mineralization Test of CPDB-Modified SA–Gelatine with Osteosarcoma Cells Incubated in Fatal Bovine Serum and Air for 7 and 14 Days, Respectively

For the mineralization test, $8 \times 10^6$ Saos-2 osteosarcoma cells were kindly provided by the Bioengineering Institute of Technology at the International University of Catalunya (UIC). The cell culture was prepared following the reference of the supplier ATCC (USA) and the method of [78] in DMEM containing McCoy's 5 medium supplement with 15% FBS and 1% penicillin/streptomycin (Sigma-Aldrich. St. Louis, MO, United States). Cells were cultivated in T25 flasks (Nunc, USA) in a humidified incubator at 37 °C, using a standard mixture of 95% air and 5% $CO_2$. After trypsinization, the cells in suspension were morphometrically analyzed. A phase-contrast microscope was used to measure the area and diameter of attached cells (n = 100) by using an image analysis system Eclipse TS2 (Nikon, NY 11747-3064, U.S.A). Detached cells were analyzed by staining 10 µL of the cell suspension with 10 µL Trepan Blue solution. The cells were then counted with a Neuberger chamber. For cell proliferation analysis, Saos-2 cells were seeded at $2.5 \times 10^5$ cells/T25 flask. On days two, four, eight, and fourteen, cells were detached, and their number was determined following the previous method. All assays were performed in triplicate, using the early log-phase to calculate the population replication time. The following procedures were followed first to detach the SaOs-2 cells to prepare them for encapsulation; old cell culture media were removed, and the cells were washed twice with 5–10 mL PBS to remove any residues from the old media; 5–10 mL Trypsin was added to the cells before incubation for 5 min at 37 °C to detach the cells. After incubation, the cells in Trypsin were centrifuged for 5 min at $300 \times g$, and then, 500 µL of McCoy medium was added to the three Falcon flasks to suspend the cells. Next, 20 µL of cell suspension was added to 80 µL of Trepan Blue to be counted by the Neuberger chamber. After counting the cells in each Falcon flask, the cells were placed again in three T25 flasks, and 8.1 mL of McCoy medium was added to suspend the cells at 37 °C in each flask, respectively. The mixture was incubated for five minutes at 37 °C. All procedures were conducted in a UV sterile hood. After incubation, 5 mL of the cells with media mixture was added to 10 mL of the optimized hydrogel, and 5 mL of the bioink was loaded to a 5 mL sterile syringe and fixed into the printing head of the Felix Bio bio printer. The printing temperature was adjusted to 4 °C at the printing bed and 37 °C at the printing head (nozzle), while the printing speed was kept at 100% and the flowrate at 100% as well. The 3D digital design file was exported as STL file from Rhinoceros 3D and adjusted to printer settings using Simplify 3D software (https://www.simplify3d.com/) and then transferred to the printer's built-in software.

The bio-printed samples were cross-linked with the cross-linking solution of reacting calcium chloride and sodium carbonate. Then, the cross-linked 3D-printed bioink patches were incubated in FBS (fetal bovine serum) and ambient air for 7 and 14 days, respectively. Each sample contained three specimens that were examined by the SEM to detect the formation of hydroxyapatite and biomineralization. Similarly to the hydrogel samples, the bioink samples (Saos-2 embedded in the hydrogel) were prepared by fixing in glutaraldehyde and kept at 4 °C. Later, the samples were mounted on steel stubs with a bio-adhesive layer and a glass cover and scanned under low vacuum.

## 3. Results and Discussion

The current study aimed to attain the sustainable synthesis of a self-biomineralized material with possible applications in tissue engineering as a bone replacing material as well as in architecture as a mineralized material that hardens by time. These characteristics are ruled by these main necessities: (1) the minimized and sustainable composition of the material by limiting the components of the material while selecting the most efficient components to achieve mineralization; (2) competent mechanical properties of the developed material balancing elasticity with plasticity to offer a rigid yet elastic material that is adequate for both applications as BRM and a structural material in the built environment; (3) autonomous mineralization without cells in non-sterile conditions, which indicates the antimicrobial trait of the developed material since the developed material is designed to be used in the built environment and to be mineralized by the ambient air, which opens wide potentials for the large-scale 3D printing of sustainable architectural applications with minimum material consumption while contributing to mitigating $CO_2$ by integrating it in this mineralization reaction; (4) slow-to-moderate degradation by achieving equilibrium between the material resistance to atmospheric conditions (resistance to water, air, and microbial contamination) while being degradable by using available buffers such as, in the case of this study, phosphate-buffered saline. This would enable the customized use of the developed material for the proposed two applications as a BRM or as a building material in the built environment.

However, the topic of optimized hydrogels for bone tissue engineering employing sustainable and green processes have been proposed in a limited number of recent studies in the bone tissue engineering field. For example, Ref. [79] offered a review on cheap and sustainable polymers that are used in bone tissue engineering, including alginate. Similarly, one study [80] proposed the use of cheap and sustainable polymers such as chitin, cellulous, and starch combining with hydroxyapatite for application in bone tissue engineering materials. Lacroix, J.; etc., 2013 [81] also proposed the use of bioactive glasses as green and safe in situ scaffolds for bone tissue engineering. One study [82] proposed self-healing sustainable hydrogels for guided bone regeneration (GBR), and a few other pieces of literature proposed the use of self-mineralized materials as building material in architecture applications either from existing cementous mineralized materials based on the $CO_2$ to calcium carbonate reactions [83,84] or by inducing mineralization through a bioactive agent as bacteria or fungi [4,85]. However, the current study goes beyond the literature by not only proposing a sustainable composed hydrogel for self-mineralized material but also by providing sustainable minimized materials and processes in preparing this self-mineralized material. Moreover, the current study novelty in the proposed hydrogel/material is that it is antimicrobial and mineralizes in ambient, open air in non-sterile conditions. Furthermore, the minimized materials used in the hydrogel composition and the cheap and relatively easy processes promote its scaling to multi-scale applications ranging from bone replacement material to construction and building material, which offers for the first time this dual use of the developed material in the field of bone tissue engineering and architecture construction, which has not been reported in the literature until now, to our knowledge.

Thus, in this section, the results and discussion show the rheological properties for the developed and optimized hydrogel in two phases, including hydrogels enhanced with different concentrations of calcium phosphate dibasic, followed by the CPDB-modified SA–gelatine hydrogels enhanced with different concentrations of EGCG, revealing the best reached rheological properties corresponding to the best hydrogel composition and concentrations. This is followed by discussion of the mineralization detection results of the optimized hydrogel incubated in different conditions without embedding cells compared to the biomineralization results in the same varied conditions when embedding osteosarcoma cells in the bioink. Finally, the results of degradation duration and conditions are also detailed.

### 3.1. Rheological Properties of SA–Gelatine Hydrogel Enhanced with Different Concentrations of Calcium Phosphate Dibasic

The current study aimed to develop a sustainable, optimized, self-mineralized hydrogel from available and efficient components to synthesize a bone replacement material. Three-dimensional polymeric hydrogels have been used widely for bone tissue regeneration for their excellent biocompatibility through high water uptake, space for drug or nutrient storage, ability for cell immobilization, and creation of a microenvironment for cell culturing, allowing nutrients and metabolites to diffuse to and from the cells, as well as biodegradability, attachment, proliferation, and differentiation due to their innate structural and compositional similarities to the extracellular matrix. Hydrogel scaffolds have proven suitable mechanical properties for promoting cell attachment and tissue formation [86]. In addition, their degradation products are not toxic in vivo. [87]. However, physically cross-linked hydrogels suffer from low mechanical strength and minute changes in the external environment (pH, temperature, and ionic strength), which affects the stability of the hydrogel system. Thus, in the current study, the author designed a three-stage chemical cross-linking reaction to cross-link the used sodium alginate by gelatine, calcium phosphate, and by the cross-linking solution composed from calcium chloride and sodium carbonate since it was reported that chemically cross-linked hydrogels are mechanically stronger and more stable because the interactions between the chemical linkages are stronger than physical linkages [88].

Tuning the mechanical properties of a hydrogel plays an important role in tissue regeneration because they create and maintain space for cell proliferation [86,89]. Generally, hydrogels suffer from insufficient mechanical strength for load bearing in hard tissue implant applications, especially for skeletally mineralized elements of the body, which indicate that the implanted site has to bear the whole amount of the applied load to maintain the mechanical stability of the implant site, which is not present in most of the common hydrogel systems. Thus, this study adopted a composite hydrogel approach that incorporates a mineral-reinforcing phase within the hydrogel to enhance its mechanical properties as a bone replacement material and as a structural load-bearing material as well. In the current study, we focused on enhancing the mechanical strength and stability of one of the most popular bio composite hydrogels, which is alginate–gelatine enhanced with different concentrations of calcium phosphate dibasic.

This ideal mechanical strength is balanced with a low-to-moderate degradation rate since the degradation of hydrogel scaffolds should be adequate for enabling new tissue formation and eventual bio-integration, and this should be neither too high, which could endanger the physical structure, nor too low as to hinder tissue regeneration and integration. The degradation rate of hydrogels is controlled by tuning the type of backbone polymer and cross-linking characteristics such as the amount of cross-linker, physical or chemical type, and size of cross-linking molecules [86,89]. The current study employed a three-phase, thermally induced chemical cross-linking to ensure the stability of the hydrogel and the low degradation rate even in aqueous environment in the absence of chemical buffers to promote its application as a self-sustaining and self-mineralizing material that can be used as a building material in the built environment and as a bone replacement material in bone

tissue regeneration applications where the in vivo conditions, including body fluids, will act as the necessary buffers for its customized degradation.

The current study adopted sodium alginate, gelatine, and calcium phosphate as the main three components of the hydrogel. These materials can mimic the properties of bone. As gelatine is derived from collagen, it exhibits biocompatibility in tissue engineering application, including biodegradability, plasticity, and adhesiveness. Thanks to its excellent solubility in the aqueous medium, gelatine can be modified without any changes in the physiochemical properties, promoting the use of gelatine widely in tissue engineering scaffolds as 3D foams or hydrogels. Furthermore, gelatine contains RGD peptide sequences, which induce excellent cell proliferation and differentiation properties [90–92]. However, gelatine cannot sustain a mechanical load, which necessitates the chemical cross-linking to overcome this deficiency in its mechanical stability [93]. On the other hand, alginate can mimic the extracellular matrix, as it is a biodegradable polysaccharide, is biocompatible, and is a less toxic hydrophilic polymer that is highly suitable for drug delivery applications [94–96]. Sodium periodate is usually used to chemically modify the polymeric chain of native alginate through oxidation reactions on the $-OH$ groups at the C-2 and C-3 positions of the uronic units of sodium alginate [97] to improve its reactive properties since oxidized alginates possess many reactive groups and a faster degradation profile than alginate alone [98,99]. Thus, the integration of sodium alginate with gelatine solves the deficiencies in both of them while augmenting their advantages [100,101].

Calcium phosphates such as hydroxyapatite and calcium phosphate have osteogenic properties and can stimulate natural bone regeneration. For example, tricalcium phosphate β-TCP is used widely for hard tissue repair applications due to its osteo-inductivity; it also has a faster dissolution rate. Some studies have reported the combination of HAp, α-TCP, and β-TCP to obtain improved mechanical properties, controlled biodegradation behavior, better osteointegration response, and increased bioactivity in the hydrogel material system. However, this indicates the use of more components under complicated processes such as a dissolution–precipitation process under physiological conditions, which does not align with the sustainable and minimized processes and material consumption approach of the current study. On the other hand, calcium phosphate dibasic is a biocompatible, bioactive, and osteoconductive reagent with a sustainable release of calcium and phosphate ions by its gradual dissolution, inducing bone formation and mineralization [102]. For example, porous calcium phosphate microspheres were used as a promising bone defect filler [103,104]. Recently, the composites of biopolymers and bio-ceramics are becoming especially popular because of their role in increasing the mechanical stability of bone substitutes and facilitating the interaction between host tissue and scaffold [105]. For instance, spherical hydroxyapatite granules encapsulated into oxidized alginate–gelatine–biphasic calcium phosphate hydrogel with different concentrations up to 35 wt% granules-loaded hydrogel proved their capacity to contribute to cell migration of rat bone marrow-derived stem cells and extracellular matrix growth [103]. Moreover, their porous bio-ceramic structure permits the infiltration and enhancement of implant tissue attachment and resorbability, resulting in faster bone ingrowth due to larger surface area [106,107]. This is due to the self-cross-linking properties and good biocompatibility of the oxidized alginate–gelatine as the hydrogel matrix for the encapsulation of granules [61,108]. Another study reported the use of alginate–hydroxyapatite composites for a drug delivery system for enzymes, antibiotics, and antifungal drugs [15]. These studies proved that the presence of hydroxyapatite along with alginate increased the cell attachment in the inner parts and offered a suitable choice of scaffold for bone tissue engineering [109]. Similarly, some literature reported that mixing gelatine with hydroxyapatite is beneficial for orthopedic applications by combining the osteo-conductivity and bioactivity of hydroxyapatite and the morphological features of gelatine [110]. Hence, incorporating gelatine into alginate and hydroxyapatite, inducing a composite, enhances the crystallinity of the composite.

In the current study, hydrogel composites consisting of sodium alginate–gelatine and enhanced by different concentrations of calcium phosphate dibasic CPDB were prepared by cross-linking the gelatine and the sodium alginate as a first-stage cross-linking, followed by cross-linking by the addition of calcium phosphate dibasic, and finally, a cross-linking solution of calcium chloride and sodium carbonate. The increased interactions between the functional groups had several effects on the materials properties, physical behaviors, and formation of hydroxyapatite crystals upon mineralization. The various concentrations of calcium phosphate affected the physical, mechanical, and morphological properties and characteristics of this hydrogel. Thus, the current study employed detailed morphological and material characterizations of the developed hydrogel for printability and its rheological properties, mineralization through formation of hydroxyapatite crystals, and degradability rate.

The sol-to-gel phase transition was completed within 4 h of mixing, as confirmed by the mixture turning to white color and by applying the syringe test, which exhibited the enhanced coherency and shape retention of the viscous gel thanks to the two-step cross-linking as well as the high molecular weight of the sodium alginate, as supported by [111]. This justifies the printing temperature at 60 °C at the printing end to facilitate the flow of the sol–gel through the printing nozzle, to avoiding clogging, and to promote layer-to-layer adhesion, which is supported by [112], who adopted around a 55 °C printing temperature to achieve 3D printing of tough hydrogels with enhanced mechanical properties by employing a two-step method to prepare the hydrogel as well by mixing at a high temperature of 95 °C, then cooling to 65 °C before printing. This is also congruent with [113], who used an up to 80 °C printing temperature to print a thermally modified alginate–di-aldehyde–gelatine hydrogel with gelatine modified at 80 °C for three hours and printed in the hierarchical, complex structures of cartilage and confirmed via μCT analysis. This is contrary to older methods reported in previous literature that usually employed from 37° to 40 °C as the printing temperature of the alginate–gelatine-based hydrogels, achieving lower mechanical properties and shape retention [114].

In the current study, a thermal shock to the printed hydrogel was provided by keeping the printing bed temperature at 4 °C to regain the gelation capacity of the viscous gel. The post-printing, third cross-linking phase of the 3D-printed CPDB-modified SA–gelatine hydrogel by the prepared solution of $CaCL_2 + Na_2CO_3$ played a crucial role in enhancing the hydrogel stability and antimicrobial property since this material is developed for non-sterile condition applications as building material in the built environment. This is congruent with [115,116], who used a 30 mg/mL $CaCl_2$ solution to cross-link SA to form strong bonding. This also helped in retaining its stability for low degradability in the absence of buffer solutions to enable its dual use as a bone replacement material in tissue engineering with customized degradation rate in vivo and as a resistant building material in the built environment in the current study.

The results of the printability test of the different tested concentrations of calcium phosphate dibasic-enhanced SA–gelatine hydrogels revealed that concentrations of 5.5, 6, and 6.5% achieved the best printability of the gel in terms of viscosity, layer coherence, structural stability, and contact angle of 35° ± 3, which is ideal for a hydrogel for moderate wettability property, as reported by [117]. Moreover, the tested different hydrogels with the three varied CPDB concentrations mentioned above exhibited minor swelling post printing and pre-cross-linking with the $Na_2CO_3 + CaCl_2$ solution, especially in lower CPDB concentrations. This was due to the temperature-catalyzed cross-linking reaction between gelatine and alginate [118], followed by the second-phase cross-linking by the addition of calcium phosphate dibasic, leading to the in situ formation of the coherent hydrogel, congruent with [119,120], and then finalized by the third cross-linking with the calcium carbonate solution. However, all the developed hydrogels suffered from shrinkage post cross-linking from 1.5 to 3 mm in all the three directions of the prints (X, Y, and Z), which is almost 0.05% to 0.1% shrinkage from the original size of the full prints. However, the shrinkage effect was more pronounced in the 5.5% CPDB-enhanced

SA–gelatine hydrogel with 3 mm shrinkage from all directions, while the least recorded shrinkage was in the 6.5% followed by the 6% CPDB hydrogel with a 1.5 mm shrinkage from all directions post cross-linking. The shrinkage rate achieved in the current study beats the 3–5% shrinkage of phosphate-enhanced alginate–gelatine hydrogels post cross-linking as reported in [121]. This shrinkage achieved in the current study in the cross-linked CPDB-modified SA–gelatine hydrogel is attributed to the hydrogen bonding in the third-phase cross-linking with the calcium carbonate solution, as supported by [122]. However, the higher concentrations of CPDB in the 6% and the 6.5% limited the effect of the shrinkage post cross-linking, which was caused by the calcium carbonate in the cross-linking solution (calcium chloride + sodium carbonate) and by the cross-linking realized by the increased calcium ions in the CPDB corresponding to its increased concentration in the hydrogel composition in the second-stage cross-linking (Figure 1). This proves that in the current study, the addition of the calcium phosphate dibasic acted as an active cross-linking agent to the sodium alginate–gelatine mixture, enhancing the hydrogel's stability and coherence.

The obtained gel exhibited dense and coherent texture with low-to-moderate irregular and interconnected porous microstructure structure with $\leq 20\%$ porosity and with pore size between 20 μm to 60 μm. This is due to the thermally induced, three-stage cross-linking of the developed hydrogel adopted in the current study (Figures 2 and 4). This is within the pore size rate of 10–500 μm reported in [123] of gelatine enhanced with different concentrations of cements. This is nearly 40% less than the reported pore size in a similar calcium phosphate-enhanced alginate–gelatine hydrogel in [124], where the pores were interconnected, and the porosity ranged from 44.45 to 67.89%, with pore size ranging from 100 to 300 μm. This can be attributed to the different preparation methods as well as the higher molecular weight of the used alginate and the three-stage cross-linking process adopted in the current study. Although even higher porosity is favorable for cytocompatibility since the pores enable the circulation of air and nutrients, the pore size is relevant to the type, scale, and morphology of the embedded cells in the bioink, with moderate pore size being more cyto-compatible [125]. However, in the current study, the material coherence balance with porosity distribution and pore size is essential to deliver both designated functions: as a hard tissue bone replacement material that is load-bearing, which indicates more coherency of the gel to enable higher mineralization density in the material, and as a stand-alone self-mineralized material for structural applications in the built environment. In the current study, the morphological and rheological characterization was conducted after drying the hydrogel in the open air for 7 days to test the actual morphology of the gel planned for use in non-sterile conditions in the built environment in addition to its possible application as bone replacement material due to its anti-microbial traits that are crucial for in vivo post-operative infection control in bone replacement grafts.

The rheological characterization of the developed gel aimed to identify the hydrogel properties dependent on the different concentrations of CPDB in balancing between elasticity and plasticity and exposing and concluding the intrinsic behavior of the material itself regardless of the thickness. Therefore, three different thicknesses from the three different concentrations of 5.5, 6, and 6.5% CPDB-modified hydrogels were examined for the rheological properties for promoting the sustainability purposes of using the least amount of materials to deliver the same mechanical qualities and also proving that the rheological properties tested translate the material properties better than the geometry characteristics. This rheological identification was decided based on the G-Values: $G'$ (storage" or "elastic" modulus), $G''$ ("loss" or "plastic" modulus), and $\tan\delta = G''/G'$, which measures the elastic ($\tan\delta < 1$) or plastic ($\tan\delta > 1$) characteristics of the developed hydrogel.

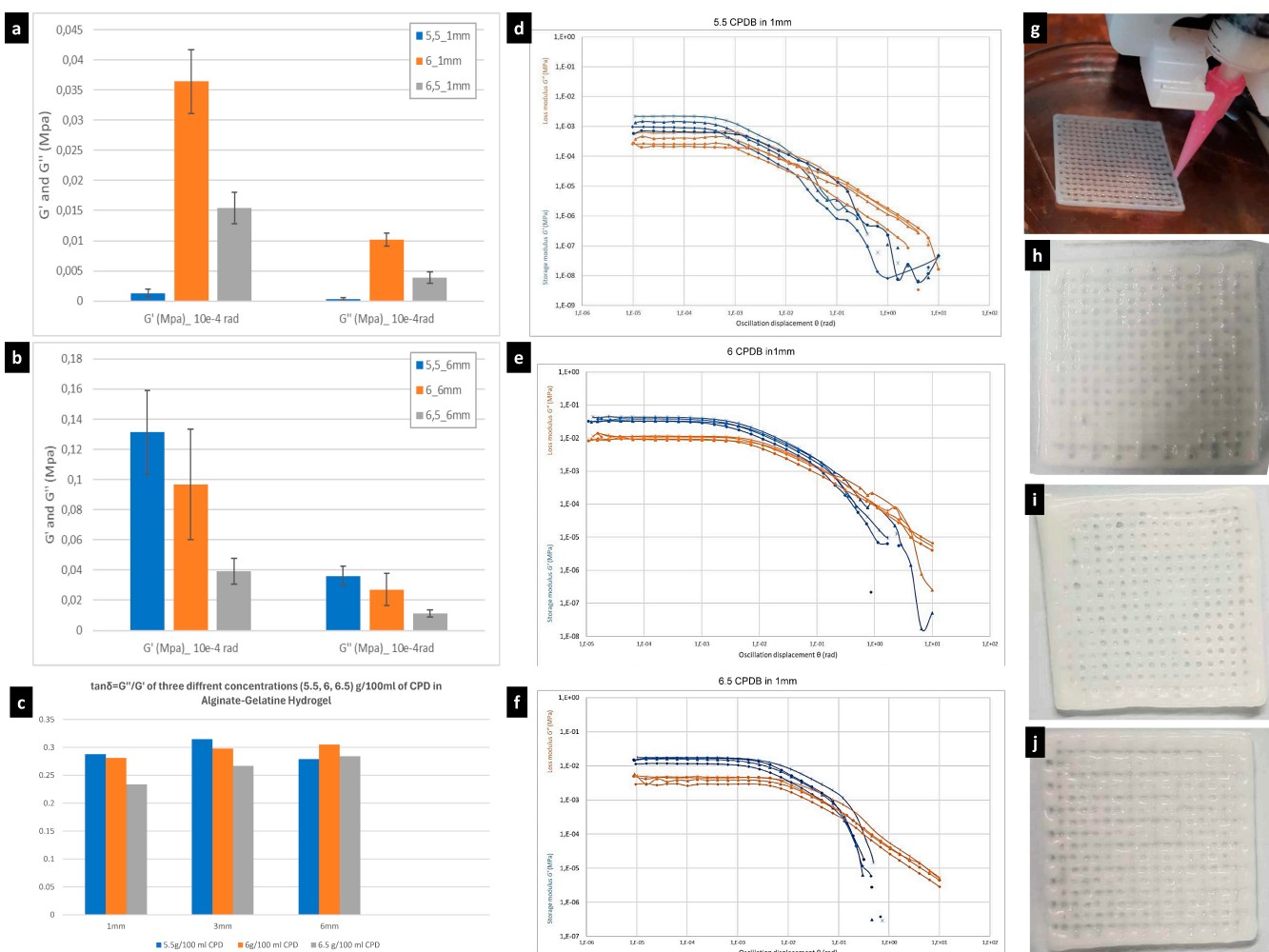

**Figure 2.** The rheological properties comparison between the three different hydrogels of different CPDB concentrations of 5.5, 6, and 6.5% enhanced SA–gelatine hydrogels in three different thicknesses of 1 mm, 3 mm, and 6 mm. (**a**) Comparison of the G values (G′ storage modulus) and G″ (loss modulus) between the three different 3D-printed hydrogels of 1 mm thickness; (**b**) comparison of G values between the three different 3D-printed hydrogels of 6 mm thickness; (**c**) a total comparison of the tan theta values of the three different hydrogels of different thicknesses of 1 mm, 3 mm, and 6 mm. (**d**–**f**) Each plot exhibits the G values (storage and loss modulus) in relation to the oscillation displacement of the tested three samples (ten replica per each): (**d**) the 5.5% CPDB rheological behavior, (**e**) the 6% CPDB properties, and (**f**) the 6.5% properties. Duplicate values obtained from overlapping in the ten replica behaviors per each sample were culled for easier reading. (**g**) The printability of the developed hydrogel of 6% CPDB-enhanced SA–gelatine hydrogel. (**h**,**j**) The 3D-printed hydrogel coherency and stability post printing and pre-third-stage cross-linking, where (**h**) corresponds to the 5.5% exhibiting swelling and low coherency; (**i**) the 6% exhibiting high coherency, layer-to-layer adhesion, and no swelling; and (**j**) the 6.5% showing minor swelling pre-cross-linking.

This indicates that the optimum criterion for deciding on the best rheological properties in the case of our study is tanδ, as it is in the middle between elasticity (tanδ < 1) and plasticity (tanδ > 1), which ensures the sufficiency of the developed hydrogel to act as a phase-transformative material, i.e., from soft to hard tissue, for application in bone tissue engineering as well as a self-mineralized material in architectural structural applications, where the initial phase of material before the mineralization is in equilibrium between elasticity and plasticity.

This justifies the selection and design of the mechanical properties' characterization of the hydrogel in the current study, where the standard rheological properties of the material expressed in its G values is the main method used to examine the behavior of the different CPDB-enhanced hydrogels [126,127]. This was done instead of testing the compressive strength as reported in other literature, where it does not reflect only the material properties but also depends to a great extent on the geometry, which is not the case in the current study that focuses in particular on studying the material itself [128].

The initial screening of the rheological properties of the different hydrogel composites with the different CPDB concentration and different thicknesses of 1, 3, and 6 mm revealed that the 6% CPDB-modified SA–gelatine hydrogel achieved this equilibrium between elasticity and plasticity in comparison to the two other best printable, coherent, and stable hydrogels of CPDB concentrations 5.5 and 6.5%. Table 1 and Figure 2a–f show the comparison between the G values and tan theta of the three different thicknesses of 1, 3, and 6 mm of the 3D-printed geometry per each concentration of 5.5, 6, and 6.5% of calcium phosphate-enhanced hydrogels.

**Table 1.** The G values and the tan theta of the tested rheological values of the three different concentrations of 5.5, 6, and 6.5 g/100 mL in three different thicknesses of 1, 3, and 6 mm each. All the results were obtained from the statistical estimation of the mean of 10 replicas per each value. The estimated standard deviation ranged from 0.003 to 0.012 for all values.

| Sample | $G'$ (MPa) $10^{-4}$ rad | $G''$ (MPa) $10^{-4}$ rad | $\tan\delta = G''/G'$ |
|---|---|---|---|
| 5.5 g/100 mL CPD in 1 mm | 0.00143814 | 0.000414447 | 0.288 |
| 6 g/100 mL CPD in 1 mm | 0.0384162 | 0.0108194 | 0.281 |
| 6.5 g/100 mL CPD in 1 mm | 0.0156726 | 0.00367397 | 0.234 |
| 5.5 g/100 mL CPD in 3 mm | 0.110418 | 0.0348274 | 0.315 |
| 6 g/100 mL CPD in 3 mm | 0.0162533 | 0.0048435 | 0.298 |
| 6.5 g/100 mL CPD in 3 mm | 0.0223588 | 0.00598952 | 0.267 |
| 5.5 g/100 mL CPD in 6 mm | 0.148103 | 0.0413324 | 0.279 |
| 6 g/100 mL CPD in 6 mm | 0.089164 | 0.027222 | 0.305 |
| 6.5 g/100 mL CPD in 6 mm | 0.0410406 | 0.0116671 | 0.284 |

As exhibited in Table 1 and Figure 2, all the tested samples of the different concentrations CPDB–SA–gelatine hydrogels exhibited a higher storage modulus $G'$ than the loss modulus $G''$ ratio, which indicates that the mixture of sodium alginate, gelatine, and calcium phosphate under the current methods is mainly elastic. This promotes its use in the designated described applications as a bone tissue replacement material and a structural material. The results also reveal an overall increase in both the storage modulus and loss modulus values congruent with the increase in the 3D prints' thicknesses. This is pronounced clearly when comparing the results between the 1 mm thickness prints of the different concentrations to the 6 mm prints of the same concentrations, respectively, although there were some fluctuations in this relativity between thickness and increase in rheological properties at the 3 mm thickness, as exhibited in Table 1, which is mainly justified by the differentiated rate of drying during the 7-day duration and the mineralization between the different concentrations, which is more augmented in this intermediate size. In the 1 mm thickness, the 6% CPDB concentration recorded the higher values of both storage modulus at 0.0384162 and loss modulus at 0.0108194 in comparison to the same thickness of 1 mm in the 5.5% and 6.5% CPDB concentrations. This might be because the 6% CPDB concentration falls in the middle between not being too-light for post drying, as in the case of the 5.5%, and not too dense for slow drying, as in the 6.5% CPDB.

However, when increasing the thickness to 3 mm, the 6% CPDB came last in the storage modulus value with 0.0162533 MPa compared to 0.110418 MPa in the 5.5% CPDB and 0.0223588 MPa in the 6.5% CPDB as well as having the smallest loss modulus of 0.0048435 MPa compared to 0.0348274 MPa in the 5.5% CPDB and 0.00598952 MPa in the 6.5% CPDB. This is justified by the uneven drying of the printed layers in the 3 mm thickness in the 6 CPDB concentration, as explained above. Nevertheless, at the 6 mm thickness, the 6% CPDB concentration recorded an intermediate storage modulus value of 0.089164 MPa between 0.148103 MPa in the case of 5.5% CPDB and 0.0410406 MPa in the case of 6.5% CPDB as well as recording the intermediate value of loss modulus of 0.027222 MPa in the case of 6% CPDB and between 0.0413324 MPa in 5.5% CPDB and 0.0116671 MPa in 6.5% CPDB.

Thus, it can be deducted that the 5.5% CPDB recorded the highest storage modulus with the 3 mm and 6 mm thicknesses while also recording the highest loss modulus in these two thicknesses, respectively. The 6% CPDB recorded the highest storage modulus in the 1 mm thickness while coming in the second place with the 6 mm thickness and third with the 3 mm thickness. Similarly, it came first in the loss modulus values with the 1 mm thickness and second with the 6 mm thickness. Finally, the 6.5% CPDB came second in the storage modulus values with the 1 mm thickness and the 3 mm while coming last with the 6 mm thickness.

Although it is obvious that increasing the thickness was congruent to the increase in both the storage and loss modulus, which reflects an overall increase in the rheological properties of the material, the main concern in the current study is the behavior of the material itself. This is why the 1 mm thickness was tested twice, and the authors thus focused on the results obtained at the 1 mm thickness to select the best CPDB concentration, which is 6% CPDB, to proceed to the next step of testing the different concentrations of EGCG as a rheological enhancer, which provides the anti-microbial effect of the developed material.

Although it was possible to compare the reached results of the developed hydrogels in the current study with the previous literature in terms of comparing swelling and shrinkage behavior, this was not the case when trying to compare the reached rheological indicators of the tested hydrogels in the current study with the previous literature because the materials and methods adopted in the current study comprise a novel, simplified, and minimized process that depends mainly on the thermal catalysis of the three-stage cross-linking process of the hydrogel developed from only three main components: sodium alginate, gelatine, and calcium phosphate dibasic, which enables its antimicrobial properties for application in built environments, as mentioned before. However, when comparing to the most relevant literature [14], it was found that a reached stiffness starting from 0.0102 MPa of GeLMA hydrogel is congruent with the reached hydrogel stiffness in the current study pertaining to the 6% CPDB-modified SA–gelatine hydrogel of 1 mm height and 0.0108194 MPa, which proves the efficiency and sustainability of the developed hydrogel in the current study in which we employed fewer materials and processes to reach to the same rheological properties of the GeLMA hydrogel reported in [14]. This is thanks to the calcium phosphate reinforcement effect [129]. Moreover, the reached rheological properties in the current study are attributed to the increased concentrations of gelatine (8%) and alginate (4%), which surpass the results obtained by [72] by employing blends of 7% alginate–8% gelatine to achieve high printability, mechanical strength, and stiffness.

### 3.2. Rheological Properties of CPDB-Modified SA–Gelatine Hydrogel Enhanced with Different Concentrations of Epigallocatechin Gallate

A polyphenolic compound, (−)-epigallocatechin-3-gallate (EGCG), which is a major catechin found in green tea, was tested in the current study as an addition to the CPDB-enhanced SA–gelatine hydrogel because of its reported health benefits, including antioxidant effects, cancer chemoprevention, improving cardiovascular health, enhancing weight loss, and protecting the skin from the damage caused by ionizing radiation. Moreover, EGCG has been shown to regulate a number of disease-specific molecular targets. These

properties give EGCG an importance in bioengineering and regenerative medicine research and applications [130]. Furthermore, it enhances the degradability of a hydrogel since it becomes rapidly degraded in both acidic (pH below 2) and neutral conditions [131,132].

EGCG, as other tea polyphenols, exhibits antibacterial properties. As was reported in the literature, epigallocatechin gallate (EGCG)'s disinfectant efficiency was measured in *Escherichia coli* cultures under different concentrations of calcium ion ($Ca^{2+}$) with reported alteration in the tested bacterial growth and structures, where higher concentrations of $Ca^{2+}$ (6–10 mM) enhanced the EGCG disinfectant effects by facilitating the entry of EGCG into the bacteria. This opens wide potentials in applying EGCG in sustainable bioactive building materials, as it can perform purifying antimicrobial effects in the built environment, and especially as an additive for the disinfection of groundwater and other raw waters containing calcium ions [133]. This also gives EGCG higher value in tissue engineering in bone replacement materials when incorporating it in calcium-based hydrogels, triggering their disinfectant effect. Applying EGCG lessens the need to incorporate additional antibiotics for infection control of the developed hydrogel grafts. For example, it was reported that (EGCG)-modified gelatine sponges incorporating β-TCP granules were applied in bone regeneration. They were prepared under vacuum-heating treatment to induce thermal cross-linking, and the prepared hydrogels exhibited high stability, β-TCP granule retention, operability, and cytocompatibility, with significant higher bone-forming ability than β-TCP alone. The treatment increased the number of osteoclasts in defects treated with the hydrogels compared with that around β-TCP alone, which indicates that the combination of EGCG, gelatine, and calcium phosphates through thermal cross-linking can be a promising process to enhance bone-forming ability [134].

This justifies the incorporation of EGCG in the current research as a disinfectant and cytocompatibility enhancer component for the SA–gelatine–CPDB hydrogel for the two proposed applications in sterile (bone replacement material) and non-sterile (building material) condition, either without or with cells.

Additionally, some research has reported the effect of EGCG in boosting biomineralization with amorphous calcium phosphate (ACP) for defect restoration by preventing and restoring demineralization, combining the antibacterial property of EGCG and the remineralization effect of ACP in both neutral and acidic conditions. In acidic conditions, EGCG exerted a strong antibacterial effect, and the ACP release rate doubled within 4 h, resulting in the prevention of demineralization in the presence of cariogenic bacteria. It also exhibited good biocompatibility with L-929 cells and human gingival fibroblasts [135].

In addition to providing support for infiltrating cells, EGCG could assure the delivery of drugs or biologically active molecules, enhancing bone formation. A study reported that developed hydrogel from gellan gum (GG), gelatine (Gel), and epigallocatechin gallate (EGCG)-loaded $CaCO_3$ microparticles is subjected to enzymatic mineralization with calcium phosphate (CaP) and recorded the release of EGCG for up to 14 days, where EGCG reduced the cytotoxicity of the calcium carbonate particles and increased the viability of the MG-63 cells exhibiting a pore size similar to that of the spongy bone when examined under optical and scanning electron microscopy [136].

Other studies have reported the use of EGCG as a safer alternative treatment for bone disorders by designing a 3D-printed tricalcium phosphate scaffold for localized EGCG delivery to enhance in vitro osteogenic ability, anti-osteoclastogenic activity, vascularization formation, and chemoprevention. It was reported that EGCG release enhances the osteogenic differentiation of hMSCs. However, it was also reported that EGCG significantly downregulates the receptor activator of nuclear factor-κB ligand (RANKL) expression by 7.0-fold, indicating that EGCG suppresses RANKL-induced osteoclast maturation as well as reducing human osteosarcoma MG-63 cell viability by 66% compared to the control at day 11. These results indicate that EGCG might be considerable only for low-load-bearing scaffolds. Nevertheless, further research on the effect of EGCG on the mineralization and its incorporation for development as load-bearing bone tissue replacement material is still

needed, especially in terms of its effect on the rheological and mechanical properties of these bone replacement scaffolds [137].

Some studies have reported the effect of EGCG on gelatine-based hydrogels' stability and mechanical behavior, indicating that the addition of EGCG improved the gel strength and thermal stability of gelatine with an optimal final concentration of 1.0 g L epigallocatechin gallate $-1$ in a gelatine solution (66.7 g L$^{-1}$) thanks to the hydrogen bonds as the main molecular interactions [138]. Moreover, some recent studies have tested the effect of EGCG on the rheological properties of active edible films prepared from tea polyphenols (TP), gelatine, and sodium alginate. They evaluated the effect of 0.4%–2.0% TP (*w/w*, TP/gelatine) on the physical, antioxidant, and morphological properties of gelatine–sodium alginate films. The results revealed that the tensile strength (Ts), contact angle (CA), and cross-linking degree were enhanced congruently with increasing TP concentration in the film, while the water vapor permeability (WVP) and light transmittance of the film were decreased. Antioxidant capacity was improved by increasing TP content in the film, and the interactions between gelatine sodium alginate and TP were confirmed by Fourier transform infrared spectroscopy analyses (FTIR). The resulting surfaces of the developed gel were smooth, continuous, and dense in internal structure as observed by scanning electron microscopy (SEM). This promotes the use of TP in gelatine and sodium alginate film solutions to improve the physical properties and antioxidant activity of the films [139].

Nevertheless, testing the effect of EGCG on the SA–gelatine–calcium phosphate hydrogel has not been sufficiently presented in the literature before, especially for load-bearing bone replacement structural material applications. Thus, in the current study, three different concentrations of EGCG were tested to evaluate their effect on the rheological properties and stability of the developed SA–gelatine–CPDB hydrogel, mainly to enhance stretchability (elasticity) and the antibacterial traits of the prepared material. The three different concentrations of EGCG, namely 4, 6, and 8 μm, were added to the optimum concentration of 6% CPDB-modified SA–gelatine hydrogel, following the recommendations of [139] for the optimum concentration of EGCG for SA–gelatine hydrogels film applications. However, the hydrogel composition used in the current study is designed for hard tissue engineering applications as a bone replacement material. This justifies reducing the tested concentrations of the EGCG for two reasons: (1) to avoid cell inhibition effects in case of embedding the osteosarcoma SaOs-2 cells in the developed hydrogel and (2) to avoid increasing the elasticity of the final material and hindering its rigidity.

The rheological tests of the three different EGCG concentrations hydrogel printed in 1 mm thick prints revealed the following: (1) All the tested concentrations resulted in reduced storage modulus and loss modulus values of the hydrogel (G′ of 0.0275944 MPa, G″ of 0.00809548 MPa in 4 μm EGCG; G′ of 0.0216016 MPa, G″ of 0.00548179 MPa in 6 μm EGCG; and G′ of 0.0277438 MPa, G″ of 0.00765127 MPa in 8 μm EGCG) in comparison to the control group (6% CPDB–SA–gelatine hydrogel without the addition of the EGCG) (G′ of 0.0384162 MPa and G″ of 0.0108194 MPa), as exhibited in Table 2 and Figure 3. Although tanδ values of the EGCG-modified gels witnessed minor changes in comparison to the non-modified gel, however, an overall reduction to the rheological properties of the hydrogel across the different tested concentrations of the EGCG is obvious, the ratio of which is ~0.001% to each of the hydrogel components' dry weight.

**Table 2.** The G values and the tan theta of the tested rheological values of the three different concentrations of 4, 6, and 8 μm of EGCG-modified hydrogel printed in 1 mm thick prints. All the results were obtained from the statistical estimation of the mean of 10 replicas per each value. The estimated standard deviation is 0.003 for all values.

| Concentration (μm) | G′ (MPa) 10$^{-4}$ rad | G″ (MPa) 10$^{-4}$ rad | tanδ = G″/G′ |
|---|---|---|---|
| 4 EGCG | 0.0275944 | 0.00809548 | 0.2933 |
| 6 EGCG | 0.0216016 | 0.00548179 | 0.2537 |
| 8 EGCG | 0.0277438 | 0.00765127 | 0.2757 |

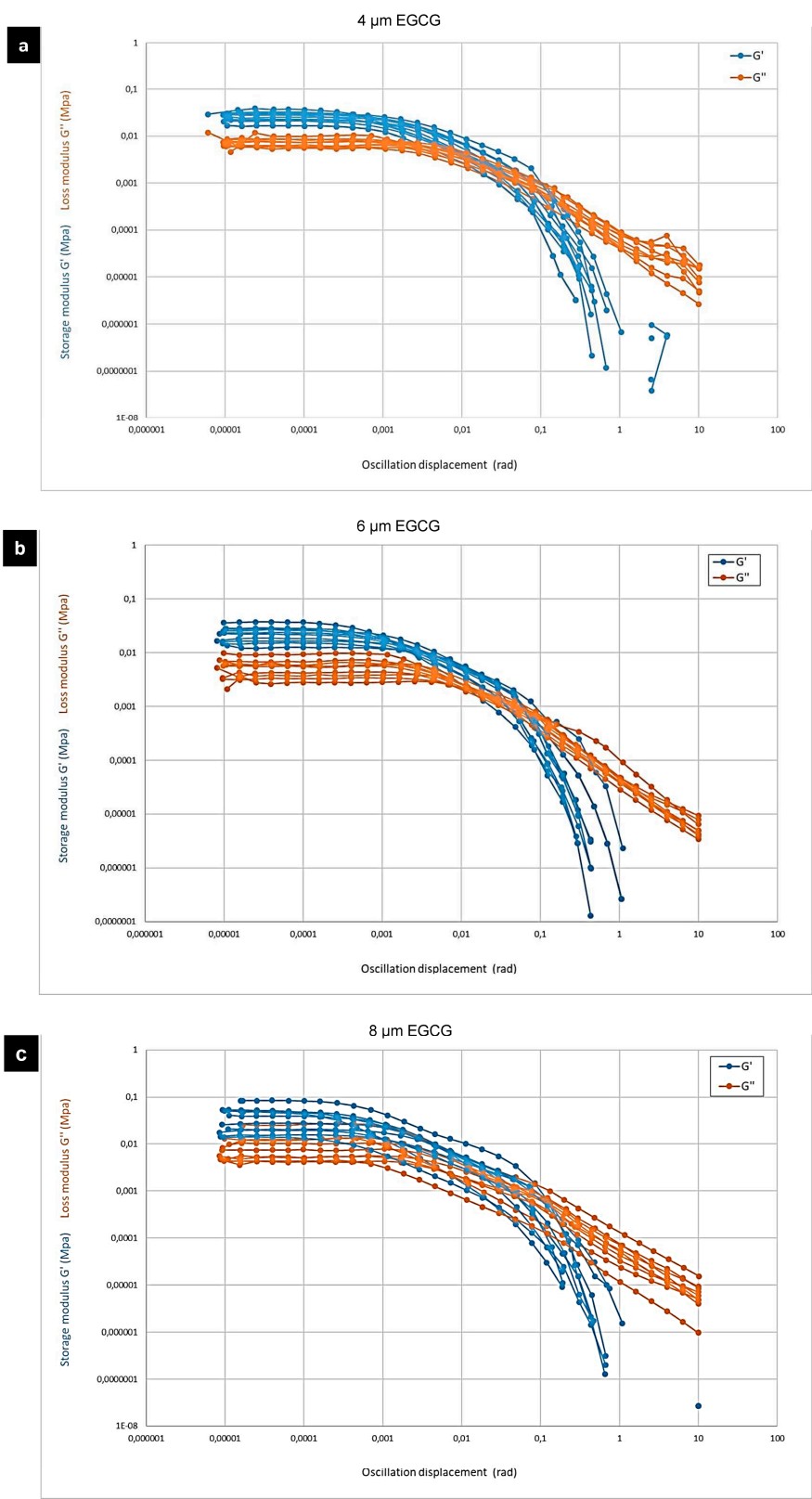

**Figure 3.** The rheological performance of the three tested EGCG-modified hydrogels using different concentrations of (4, 6, and 8 μm). Each plot exhibits the G values (storage and loss modulus) in relation to the oscillation displacement of the tested three samples (ten replicas each): (**a**) the 4 μm EGCG-modified hydrogel rheological behavior, (**b**) the 6 μm EGCG-modified hydrogel properties, and (**c**) is the 8 μm EGCG-modified hydrogel properties.

It is noteworthy that the rheological tests were always performed on the samples after 1 week of drying in ambient air (non-sterile conditions), which indicates that the EGCG-modified hydrogel either suffered from reduced mineralization due to the reduced ability to interact with the ambient $CO_2$ or that it suffered slower interaction and mineralization in the ambient air. Thus, further testing of the rheological properties of the EGCG-modified hydrogel will be conducted in the future to test a wider array of different concentrations as well as the change in their rheological properties chronologically since this would exceed the scope and capacity of the current research, which is focused on testing the effect of the EGCG of the developed hydrogel only as a possible antibacterial and stretchability booster.

The G values in all the three tested concentrations of EGCG revealed minor differences and fluctuations, which indicates that the used concentrations might have needed to be significantly varied to sense their effect or that EGCG has a trivial effect on the current hydrogel composition due to the three-stage thermally and chemically induced cross-linking, which might have affected the behavior of the EGCG.

Based on the reached results, EGCG was not found to efficiently promote the rheological properties of the developed hydrogel, especially since the rheological properties in the current study were measured after one week of drying in ambient air, which implies the interaction and mineralization of the EGCG modified hydrogel become unfavorable for the designated application as bone replacement material as well as a structural building material in non-sterile conditions. Furthermore, the EGCG as a polyphenolic tea extract compound is considered an expensive reagent, which hinders its sustainability and alignment with the current study objectives of developing an affordable and available hydrogel for multi-scale applications. Thus, the authors maintained the optimum CPDB concentration-modified SA–gelatine hydrogel for testing its mineralization and microstructure when incubated in different conditions to measure the feasibility of applying the material both inside the human body as a bone replacement material as well as a building material self-mineralized in non-sterile conditions.

### 3.3. Mineralization of the 6% CPDB-Modified SA–Gelatine Hydrogel Incubated in Fetal Bovine Serum and Air

The design of this experimental procedure was based on the objective of the multi-scale application of the developed material as a load-bearing, self-mineralized, bone replacement material and as a structural building material that can be adopted in open-air, non-sterile conditions. This indicates that the main criteria is the mineralization of this CPDB-modified SA–gelatine hydrogel by detecting the formation of hydroxyapatite $[Ca_{10}(PO_4)_6(OH)_2]$ crystals, which is the characteristic mineral phase of bones, on the 1 mm thick prints of the developed hydrogel.

In the design of the experimental procedures in this phase, we chose incubation in FBS and air, respectively, due to the following reasons; (1) Fetal bovine serum (FBS) is a common media constituent used in tissue engineering cultures [140] thanks to its nourishing composition containing proteins and growth factors, allowing maintenance and proliferation of eukaryote cells in vitro It has proven to maintain and boost the growth of different type of cells, which justifies its standard use in cell culture and tissue engineering protocols [141]. Although in the current phase of the experimental study, the optimized hydrogel does not contain any embedded osteosarcoma cells as planned, the use of FBS for incubating the hydrogel was intended not only to standardize the experimental study procedures by providing a control group where the hydrogel is incubated in FBS without embedding cells but also to test the effect of the FBS as a standard simulation of body fluids [142,143] on the degradation rate and mineralization behavior of the optimized hydrogel in the current study. (2) Incubation in open-air, non-sterile conditions is considered the pivotal point of novelty in the current study since it has not been reported in the previous literature before, especially for the novel proposed application in the current study as a structural building material, to our knowledge. The standard tissue engineering practices have always focused on in vivo testing of the developed hydrogels scaffolds,

which necessitates their sterilization. This is not the case in the current study, which takes a different approach to prove that under non-sterile conditions, the incubated, optimized hydrogel exhibits high antimicrobial properties, as exhibited in Figure 4, which shows the SEM results of the incubated, optimized hydrogel in open-air, non-sterile conditions.

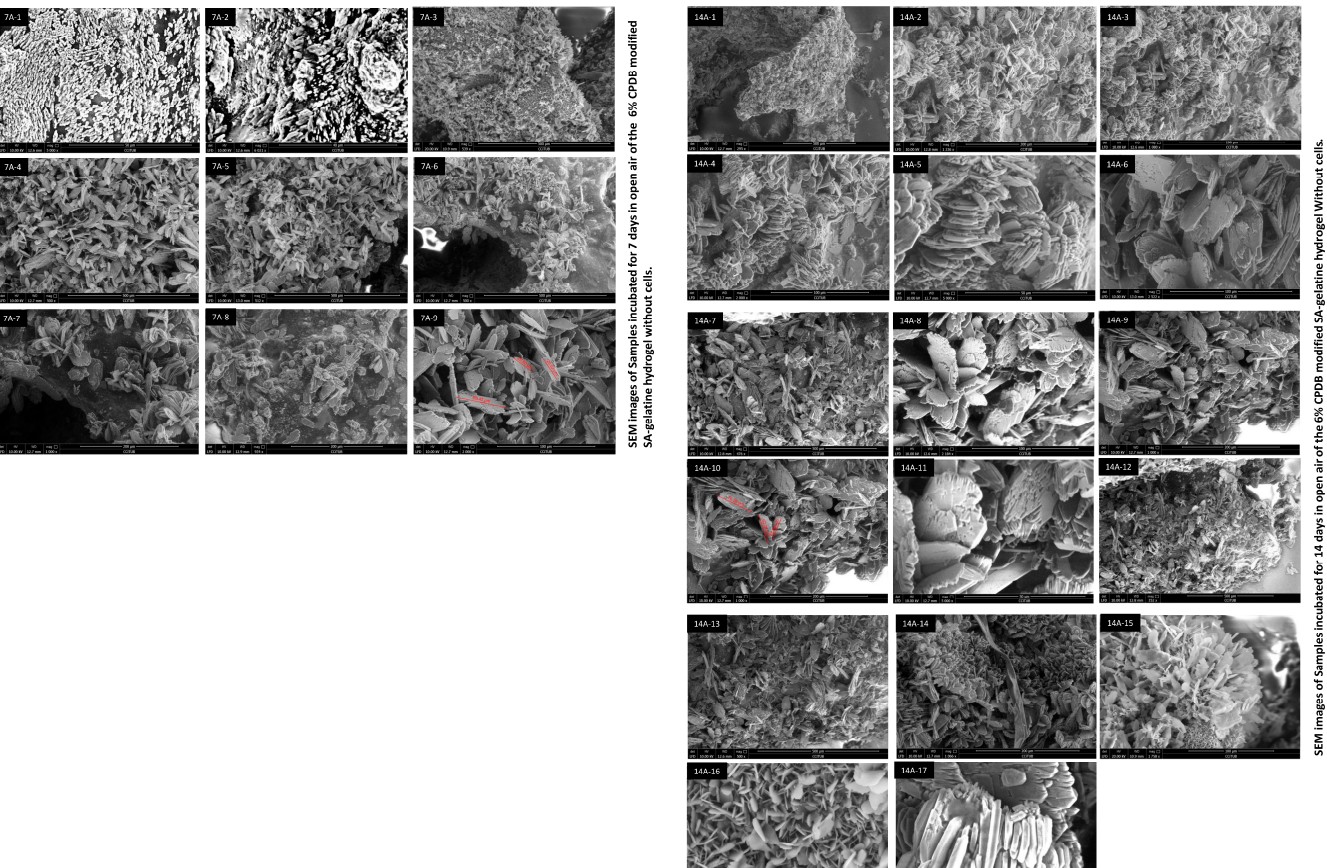

**Figure 4.** Scanning electron microscopy results of the optimized hydrogel samples (without cells) incubated in open-air, non-sterile conditions for 7 and 14 days, respectively. (7A-1 to 7A-9) The mineralization detection results of the first group of the optimized hydrogel samples incubated in air for 7 days, where obvious and dense formation of hydroxyapatite platelets is manifested in all the SEM images. (14A-1 to 14A-17) The second group of the scanned hydrogel samples incubated in air for 14 days, where all the samples exhibit a significant increase in the density, projection, and orientation of the hydroxyapatite platelets, proving the mineralization of the developed optimized hydrogel in open-air, non-sterile conditions.

The detection of the formation of the hydroxyapatite crystals was conducted by scanning electron microscopy (SEM) on the surface of the hydrogel, using the low-vacuum mood method without further fixative solutions to avoid damaging the samples or altering their chemical structure, as congruent with the methods of [144,145]. This is justified because the surface of the 6% CPDB-modified SA–gelatine hydrogel is a reactive interface in each of the cases, i.e., either in FBS or with the surrounding air containing $CO_2$ and $O_2$, which induces the mineralization of the hydrogel. A similar green approach of low cost and low environmental impact material for environmental applications and medical use that induces the formation of hydroxyapatite was tackled by [146]. However, it was conducted in aqueous media, where the synthesis of the hydroxyapatite was achieved by reacting phosphoric acid with calcium carbonate in a water suspension to form a Ca–HA gel of fine particles. Moreover, another recent study [147] proposed using a catalyst of an active hydroxyapatite (HAp) and brushite (Bru) mixture to fixate carbon dioxide from

the ambient air as a green approach to produce formic acid or acetic acid as well as to alleviate the carbon dioxide foot print. However, in the current study, we go beyond the current literature by providing a self-mineralized material in ambient air that hardens chronologically, depending on the available minimized components and using thermally induced chemical reactions from the outer layer of the material as the reactive interface to produce hydroxyapatite on the surface of the hydrogel while maintaining the elasticity of the interactive material in its inner layers, following an exact biomimetic approach of the bone formation and mineralization process, i.e., following the anatomical level of long bone (e.g., femur), where the harder, mineralized layer of the bone is the outer layer (cortical bone), while moving inwardly, there is less dense and more of cancellous bone [148,149].

The reaction cycle of hydroxyapatite calcination in native mammalian bone tissue was illustrated in an equation presented in [150], where organic matter, hydroxyapatite, and water, through the calcination phase, turned to calcium phosphate and carbon dioxide. In the current study, we propose a reaction of mineralization of the trio-component hydrogel of sodium alginate, gelatine, and calcium phosphate dibasic cross-linked with calcium chloride and sodium carbonate to formulate hydroxyapatite crystals. This approach is supported by an increasing number of recent studies mainly focused on the synthesis of hydroxyapatite [151]. The standard synthesis of hydroxyapatite is usually achieved by reacting soluble calcium salts [152]. Also, different sources of calcium in the solid state can improve the reaction as well [153]. However, such a reaction is not a simple, straightforward reaction but rather a group of sequent complex reactions, especially in the current study, with three stages of thermally induced chemical cross-linking and incubation in open air. However, the following equation (Equation (1)) draws an overall view of the proposed mineralization reaction, exhibiting the chemical reactive groups in the input components of the hydrogel and the outputs mainly exhibiting the hydroxyapatite crystals, confirming the mineralization of the hydrogel with some expected trace elements of Na, N, and Cl ions that can be present as salts, for example, NaCl.

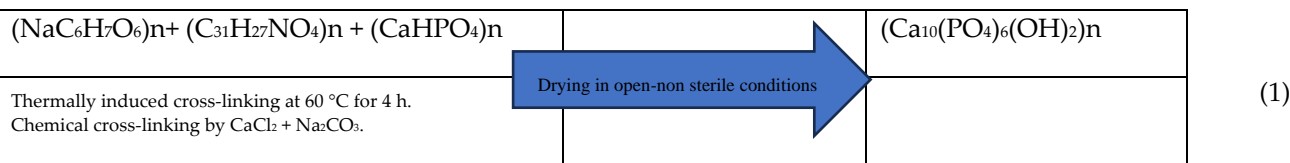

$$(NaC_6H_7O_6)n + (C_{31}H_{27}NO_4)n + (CaHPO_4)n \xrightarrow{\text{Drying in open-non sterile conditions}} (Ca_{10}(PO_4)_6(OH)_2)n \tag{1}$$

Thermally induced cross-linking at 60 °C for 4 h.
Chemical cross-linking by $CaCl_2 + Na_2CO_3$.

Equation (1): The proposed reaction of hydroxyapatite formation from mineralization of the three-stage, thermally induced, chemical cross-linked hydrogel of 6% CPDB-modified SA–gelatine by incubation in open air ($CO_2$, $O_2$) and in non-sterile conditions. The equation exhibits the chemical composition of the hydrogel developed from sodium alginate, gelatine, and calcium phosphate dibasic, cross-linked thermally and chemically by the cross-linking solution prepared from reacting calcium chloride and sodium carbonate. The mineralization occurs after incubating the hydrogel in open air for 7 and 14 days, respectively.

The detected mineralization of the air-incubated hydrogel samples obviously proved the formation of hydroxyapatite platelets in all the SEM-analyzed samples, as exhibited in Figure 4 congruent with [152], showing the two groups of tested samples of the optimized hydrogel of 6% CPDB–SA–gelatine incubated in open-air, non-sterile conditions for 7 days and for 14 days, respectively. The figure shows that the density, size, and spatial extension of the hydroxyapatite crystals increased significantly to fill all the hydrogel interfaces (14A-1–14A-6,14A-14–14A-17), with scale bar ranging from 500 μm to 30 μm, as in Figure 4 (14A-17). The first group of SEM images in Figure 4 exhibit the results of the mineralized samples of the optimized hydrogel incubated for 7 days in open air, where the samples were still in the active mineralization process, and some empty regions in the hydrogel interface can be detected, as exhibited in Figure 4 (7A-1,7A-2,7A-6,7A-7,7A-8). And the overall lower density of the hydroxyapatite platelets in this first group (7-day air incubation) is pronounced along the different magnifications used, ranging from 500 μm to 30 μm scale bar, when compared to the second group's mineralization density (14-days air

incubation). This can be seen when comparing (7A-1 to 7A-9) scans from the first group with (14A-1 to 14A-17) in the second group. This is mainly justified by the drying of the hydrogel in the open air in room temperature, where it loses its water content, almost reversing the first calcination equation proposed by [150], which is congruent with the chemical reactions involved during the calcination and precipitation methods exhibited in in the same reference [150]. However, the current study employed lower temperature in initiating the mineralization reaction, instead adopting chemical pathways where breaking the calcium carbonate (3rd-stage cross-linking) occurred as a result of adjusting the reactive groups between the hydrogel interface and the surrounding air, as exhibited in Equation (1). Furthermore, the presence of calcium carbonate facilitated the mineralization reaction and the formation of the hydroxyapatite by providing more reactive sites between the hydrogel interface containing the calcium carbonate ($CaCO_3$) and water ($H_2O$) and the surrounding air containing $CO_2$ and $O_2$ [4,150] to facilitate the hydroxyapatite platelets formation. Although the $CaCO_3$ catalyzed the mineralization by air reaction in the current study, the formation of the detected hydroxyapatite crystals is attributed mainly to the high concentrations of the calcium phosphate dibasic in the reaction, as supported by [154], who proved the self-setting and hardening of calcium phosphate cements and their ability for forming hydroxyapatite crystals. Furthermore, the formation of the hydroxyapatite platelet morphology proved by the SEM scans in the current study is congruent with [150,155], who proved the transformation of calcium phosphate particles to platelet-like apatite crystals.

Unlike the mineralization morphology and density detected in the optimized hydrogel samples (without cells) incubated in FBS (Figure 5), the identical hydroxyapatite platelet morphology is not visible in the scanned hydrogel interface, but rather, multiple irregular, varied-sized faceted spheres are detected in all the scanned hydrogel samples incubated in FBS for 7 or 14 days. This still signifies the presence of hydroxyapatite; however, in another morphology of faceted spheres, as reported in [156], the SEM images of the encrusted crystals in artificial urine solutions (AUS) exhibited hydroxyapatite spheroid-like structured crystals observed in the calcium phosphate/struvite AUS. This proves that the detected faceted spheres in the SEM images of the hydrogel incubated in fetal bovine serum in the current study (Figure 5) are of hydroxyapatite since both FBS and urine are a simulation of body fluids. Moreover, Refs. [150,157], proved the varied scales and shapes of faceted, spherical hydroxyapatite crystals using scanning electron microscopy. Furthermore, the varied morphology of the hydroxyapatite crystals between incubation in air (sharp-edged platelets) and incubation in FBS (faceted spheres) is justified by the different chemical structure of the media that affected the crystalline structure of the formed hydroxyapatite [150,156]. The fetal bovine serum is a rich source of mineral ions such as sodium, chlorine, potassium, calcium, and phosphate as well as proteins such as albumin, bilirubin, creatinine, and globulin and lipids such as cholesterol and monosaccharides such as glucose [158,159]. This vivid chemical composition enables more complex reaction pathways that result in the faceted sphere form of the detected hydroxyapatite crystals in the scanned samples of the hydrogel incubated in FBS, as exhibited in Figure 5.

However, the density of these faceted spheres of hydroxyapatite and the overall texture of the hydrogel surface proves the reduced mineralization of the hydrogel incubated in FBS. This was more pronounced in the 7-day period of incubation in FBS, where 7 FBS-2, 7 FBS-4, and 7 FBS-8 yielded many areas of the hydrogel surface without any detected hydroxyapatite crystals, even in the faceted spheres morphology, while the projection (textures) of the second group of the hydrogel samples incubated in FBS for 14 days exhibited increased density of the faceted hydroxyapatite spheres, as exhibited in Figure 5 (14FBS-1,14 FBS-4). This proves that the longer incubation period in FBS facilitated interaction with the hydrogel interface to produce more hydroxyapatite crystals.

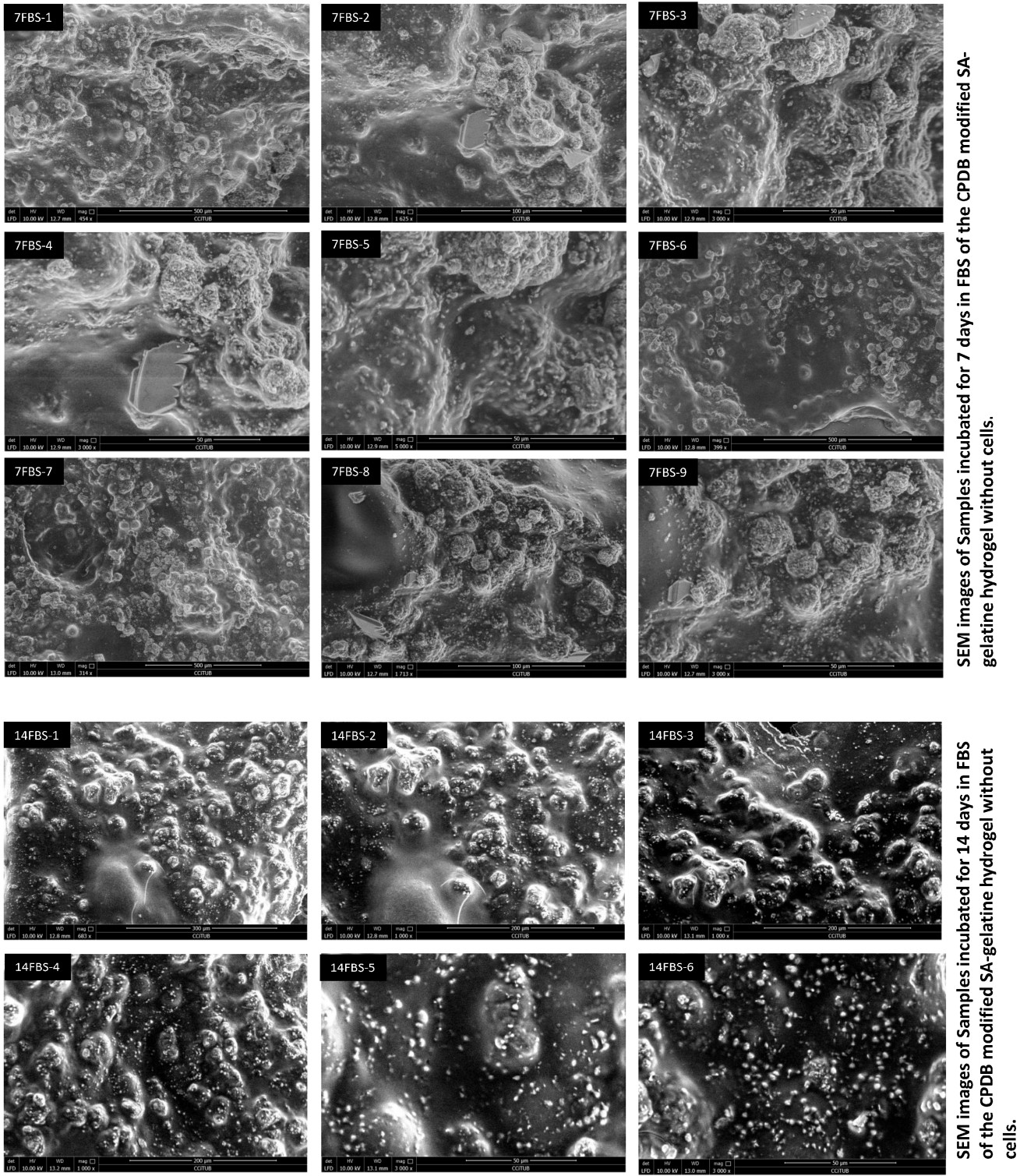

**Figure 5.** Scanning electron microscopy results of the optimized hydrogel samples (without cells) incubated in FBS for 7 and 14 days, respectively. (7 FBS-1 to 7 FBS-9) First group of the optimized hydrogel incubated in fetal bovine serum for 7 days, with different magnification ranging from 50 to 500 μm scale bars, exhibiting the formation of hydroxyapatite crystals with a faceted spheres morphology. (14 FBS-1 to 14 FBS-6) SEM results of the second group of samples of the optimized hydrogel incubated in FBS for 14 days, with different scalebars ranging from 50 to 200 μm, which exhibits enhanced texture of the mineralized hydrogel with increased density and projection of the faceted hydroxyapatite spheres.

Comparing the results of the hydroxyapatite density, morphology, and distribution, over all the developed hydrogel surfaces between the samples incubated in air and the samples incubated in FBS, it can be concluded that the incubation in air was more potent in accelerating the mineralization of the samples and densifying the resultant hydroxyapatite platelets across the topology of the SEM-scanned hydrogel samples.

### 3.4. Mineralization Test of CPDB-Modified SA–Gelatine Encapsulating Osteosarcoma Cells Incubated in Fatal Bovine Serum and Air for 7 and 14 Days, Respectively

The SEM results of the hydrogel samples incubated in air and in FBS, respectively, proved the capacity of the proposed optimized hydrogel and the three-stage cross-linking process in achieving mineralization of the hydrogel and forming hydroxyapatite crystals in various media. It was also mandatory to detect the mineralization capacity of the hydrogel with the embedded osteosarcoma cells within it (the bioink). In the current study, the main objective was to prove the hydrogel's capacity without cells to achieve self-mineralization, especially in air, as the main sustainable approach of developing this multi-scale application material. On the other hand, the capacity of the bioink including the embedded osteosarcoma SaOs-2 cells to achieve mineralization came second in importance in the current study. This is because the employment of SaOs-2 cells, culturing, and maintenance in bone replacement material is not a sustainable approach for two reasons: (1) An optimized hydrogel (without cells) applied in bone tissue engineering scaffolds or grafts is designated to be applied in vivo, where the living, healthy cells responsible for bone mineralization are already existing (patient cells), which indicates there is no compatibility to use the osteosarcoma SaOs-2 cell lines for embedding in the hydrogel as bone replacement material for active mineralization and bone remodeling; (2) usually, testing the effect of the hydrogel on cell viability in the literature is conducted for short periods (21 days) due to the sensitivity of the cell culture and the complex maintenance procedures that it requires, added to the complex and inaccurate statistically based estimations of the toxicity tests, which give only an approximated idea of how the cells act under the developed hydrogel. Therefore, in the current research, the authors designed the experimental study to detect the mineralization of the hydrogel under the effect of embedded osteosarcoma cells as an indicator of the cells' viability, bioactivity, and achieved metabolic pathways to realize the mineralization, aligning with the objective and methodology of the current study of developing optimized hydrogel with sustainability in its composition (materials) and operations (processes). Thus, the mineralization of the bioink (optimized hydrogel+ the osteosarcoma SaOs-2 cells) was detected using scanning electron microscopy of the samples of the bioink incubated in air in controlled but non-sterile conditions or incubated in FBS at 37 °C.

In the current study, the osteosarcoma SaOs-2 cell line was used thanks to its availability and reliable reproducibility in unlimited numbers without the need for isolation or ethical approval. The human osteosarcoma cell line SaOs-2 is a phenotypically mature osteoblast with high levels of ALP activity, exceeding other osteosarcoma cell lines such as MG-63 and SaOs-1. These characteristics serve synthesizing autonomous biomineralized materials with greater spatial dimensions and viability. Additionally, it was reported that the ALP activity by the SaOs-2 cell line was similar to human primary osteoblasts at early time points but achieved 120-fold higher results after 14 days of culturing under the same conditions [160]. The capacity of SaOs-2 cells to form a calcified matrix typical of woven bone was also reported [161] (Saldaña, et al., 2011) as well as the high-detailed similarity between the collagen structure synthesized by SaOs-2 and the collagen formed by primary human osteoblast cells but with a higher level of lysyl hydroxylation in the SaOs-2 cells [162]. Finally, the similarity between the cytokine and growth factor expression of SaOs-2 cells and primary normal human osteoblast cells was also reported [163,164]. The SaOs-2cells were employed in the current work due to their strong similarity to human isolated osteoblasts, giving the possibility of application on a wide spatial scale due to its enhanced ability of proliferation. This was proven after only 14 days of incubation, achiev-

ing an $8 \times 10^6$ cell count/T25 flask using the Neuberger chamber. The bioink composed from the optimized hydrogel and the cell culture was printed in cell-sustaining conditions of 37 °C and 100% printing speed and flow to minimize the post-printing pre-cross-linking time. This is congruent with [165]. After cross-linking, the 3D-printed bioink samples were incubated in FBS at 37 °C as a simulation of the in vivo conditions of using these developed grafts as a bone replacement material [114,166]. Usually, in the literature, a toxicity test is conducted to measure the cell viability under the effect of the developed hydrogel, which is usually conducted by immunofluorescence microscopy methods that employ cell staining as a differentiating marker between the living and the dead cells [167–169]. However, these methods imply exposing the cells to the toxicity effect of the stains and radiation used in these procedures, which hinders the sustainability of the procedure and reduces the accurate prediction of the hydrogel toxicity and the viability of cells under its effect. Thus, the authors adopted a sustainable approach in the current study to conclude the effect of the hydrogel on the cells' viability and behavior through measuring the osteosarcoma cells' effect on the mineralization process of the hydrogel (Figure 6) and comparing it to the control group, which is the hydrogel incubated in FBS (without cells) (Figure 5). The detection of the mineralization of the bioink incubated in FBS was conducted by fixing the samples first, then scanning them under low-vacuum scanning electron microscopy, as discussed before.

The results of the SEM revealed enhanced mineralization of the bioink (the hydrogel with the embedded osteosarcoma cells) in terms of density and the size of the formed faceted, spherical hydroxyapatite crystals, as exhibited in Figure 6 (7FBS-SaOs-2-1–7 FBS-SaOs-2-5), in comparison to their density, size, and distribution in the case of the hydrogel without cells incubated in fetal bovine serum for 7 days (Figure 5 (7FBS-1–7FBS-9)). This is attributed to the bioactivity of the osteosarcoma cells and their metabolic pathways being incubated in FBS, which induced the mineralization of the material and the composition of the hydroxyapatite crystals with larger size, density, and distribution along the bioink prints' topology. Similarly, a mild increase in the density of the formed hydroxyapatite crystals was detected in the bioink prints incubated in FBS for 14 days, as exhibited in Figure 6 (14 FBS-SaOs-2-1–14 FBS-SaOs-2-7), in comparison to the control group (hydrogel without cells incubated in FBS for 14 days) (Figure 5 (14 FBS-1–14 FBS-6)). This proves that the hydrogel has cytocompatibility, by encapsulating, and maintaining the osteosarcoma SaOs-2 cells' viability and enabling their bioactivity in inducing mineralization and hydroxyapatite formation.

The role of the SaOs-2 cells in inducing mineralization was further confirmed by analyzing the SEM results of the bioink prints incubated in air in controlled but non-sterile conditions for 7 and 14 days, respectively, as exhibited in Figure 7. As shown in both groups of SEM images, the first group (from 7Air-SaOs-2-1 to 7Air-SaOs-2-9) incubated in air for 7 days and the second (from 14Air-SaOs-2-1 to 14Air-SaOs-2-9) incubated for 14 days exhibit significantly increased density, distribution, sharpness, and projection of the formed faceted hydroxyapatite spheres, starting at crystal of 2 µm size, when comparing them to the bioink samples (with cells) incubated in FBS (Figure 6). Although this proves the bioactivity of the osteosarcoma cells in boosting the mineralization of the hydrogel incubated in air as well as the biocompatibility of the developed hydrogel for maintaining cell viability and boosting its pathways, when comparing these results (Figure 7) with the control group of the mineralized hydrogel samples incubated in air without cells (Figure 4), it can be inferred that the calcium phosphate dibasic's reaction with air as exhibited in Equation (1) is more responsible for the achieved results of the dense mineralization of the bioink incubated in air (Figure 7). This is due to two reasons: (1) The exhibited well-defined and sharp morphology of the faceted hydroxyapatite spheres in the bioink incubated in air is sharper and denser than the bioink incubated in FBS (Figure 6), which proves the mineralization and hardening of the hydrogel's outer interface that is encapsulating (coating the embedded cells) by the reaction with air, as described above in Section 3.3, although incubating the bioink with the living cells is more compatible with FBS to maintain

motile conditions for cells' maintenance and circulation of nutrients and oxygen [170]. This proves that incubating the hydrogel and the bioink with embedded cells in air induces higher mineralization and formation of the hydroxyapatite crystals. (2) This is thanks to the high concentration of calcium phosphate dibasic in the developed hydrogel, where in addition to its biocompatibility, bioactivity, and osteoconductive properties [171], it gradually dissolves to release calcium and phosphate ions, which are beneficial in the hydroxyapatite formation [172]. The dissolution of the calcium phosphate depends on the surface exposed to the physiological environment and cellular activity. Therefore, the mineralization of the bioink (with the embedded cells) incubated in air was denser than the mineralization of the bioink incubated in FBS, where the particles of the calcium phosphate embedded inside the hydrogel will slowly interact with the environment at the outer peripheral surfaces as the hydrogel degrades, showing that when incubating the bioink samples in air, the outer surface reacts faster with the surrounding carbon dioxide and oxygen from the ambient air to form the hydroxyapatite crystals, with slower degradation of the hydrogel matrix that is reinforced by the calcium phosphate dibasic. This is unlike incubation in FBS, where a slower interaction rate and faster degradation rate of the hydrogel occurs, leading to a slower mineralization rate and lower density of the hydroxyapatite formation [124]. Thus, it can be concluded that the optimized hydrogel is the 6% CPDB-modified SA–gelatine mineralizes in open air with anti-microbial properties, as exhibited in the SEM images of Figures 4 and 7, both with and without cells. And it mineralizes with a slower rate in FBS either with cells or without cells, with more detected mineralization in the case of the embedded osteosarcoma cells, which proves the compatibility of the hydrogel with the SaOs-2 cells since they together boosted the mineralization of the hydrogel and the formation of the hydroxyapatite crystals. The effect of the mineralized hydrogel on the embedded osteosarcoma cells enabled maintaining the cells incapsulated within the mineralized outer interface of the hydrogel, isolating them from the open-air conditions yet still enabling their access to oxygen and nutrients circulation gained either from the original carrier media of the SaOs-2 cells added to the hydrogel or from the mineralization reaction products from the outer to the inner layers of the hydrogel. A further detailed analysis of the mineralization reaction kinetics of the bioink and the reciprocal effect of the hydrogel on the cell viability and proliferation will be exhibited in future studies due to the tight scope and objective of the current work.

The degradability of the hydrogel was evaluated by measuring the increase in its dimensions after the two weeks of incubation in FBS, without or with cells, as a function of incubation time in FBS at 37 °C. The authors chose to test the degradation of the hydrogel under the same media of incubation for the following reasons: (1) FBS is a closer simulation to body fluids, where the proposed application of the optimized hydrogel will be conducted as a bone replacement material (graft or scaffold); (2) FBS is used to sustain 3D cell cultures of different bone cell lines, including the SaOs-2 cells used in the current study; and (3) lastly, the test maintains the sustainable approach of developing the optimized hydrogel for multi-scale applications, as mentioned throughout the manuscript. The degradation process occurs initially by breaking a small number of cross-linked bonds, which cannot damage the whole hydrogel network but rather increases the lattice size of the networks, resulting in a large water absorbance [173] until reaching a critical value where the whole cross-linking network dissociates, leading to the loss of hydrogel. After incubation in FBS for 14 days, all the hydrogel and bioink samples displayed an increase in the dimensions of the prints with a 5 to 20% ratio, with higher increase in dimensions in the case of bioink (hydrogel with embedded cells). This can be justified because the viscosity of the hydrogel is already diluted by the effect of embedding the SaOs-2 cells with their liquid media in the hydrogel. In other words, the addition of the cells with their medium already dilutes the concentration of the hydrogel and affects its cross-linking network, leading to an increased swelling effect and subsequent increase in dimensions. However, this reached degradation rate after 14 days of incubation in FBS renewed every 2 days is a promising result since a

hydrogel should be maintained for a suitable time in order to enable regeneration of new tissue in case of application as a bone graft or bone replacement material.

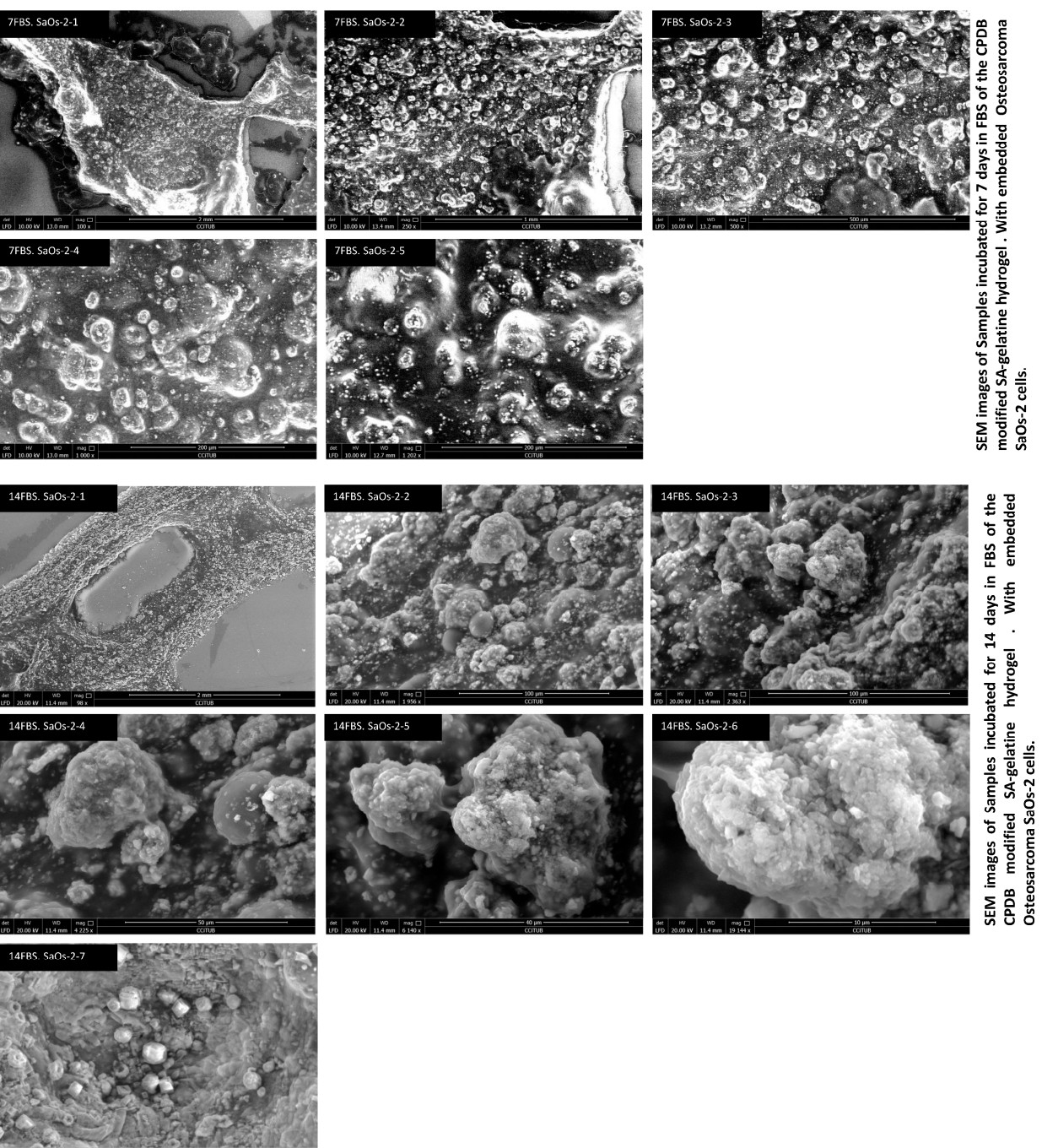

**Figure 6.** SEM images of the detected mineralization of the bioink prints (optimized hydrogel of 6% CPDB-modified SA–gelatine with embedded osteosarcoma SaOs-2 cells) incubated in FBS for 7 and 14 days, respectively, at 37 °C. (7FBS-SaOs-2-1 to 7FBS-SaOs-2-5) The first group of the SEM images of the scanned bioink prints samples incubated in FBS for 7 days. (14 FBS-SaOs-2-1 to 14 FBS-SaOs-2-7) The second group of the SEM images of the scanned bioink prints incubated in FBS for 14 days.

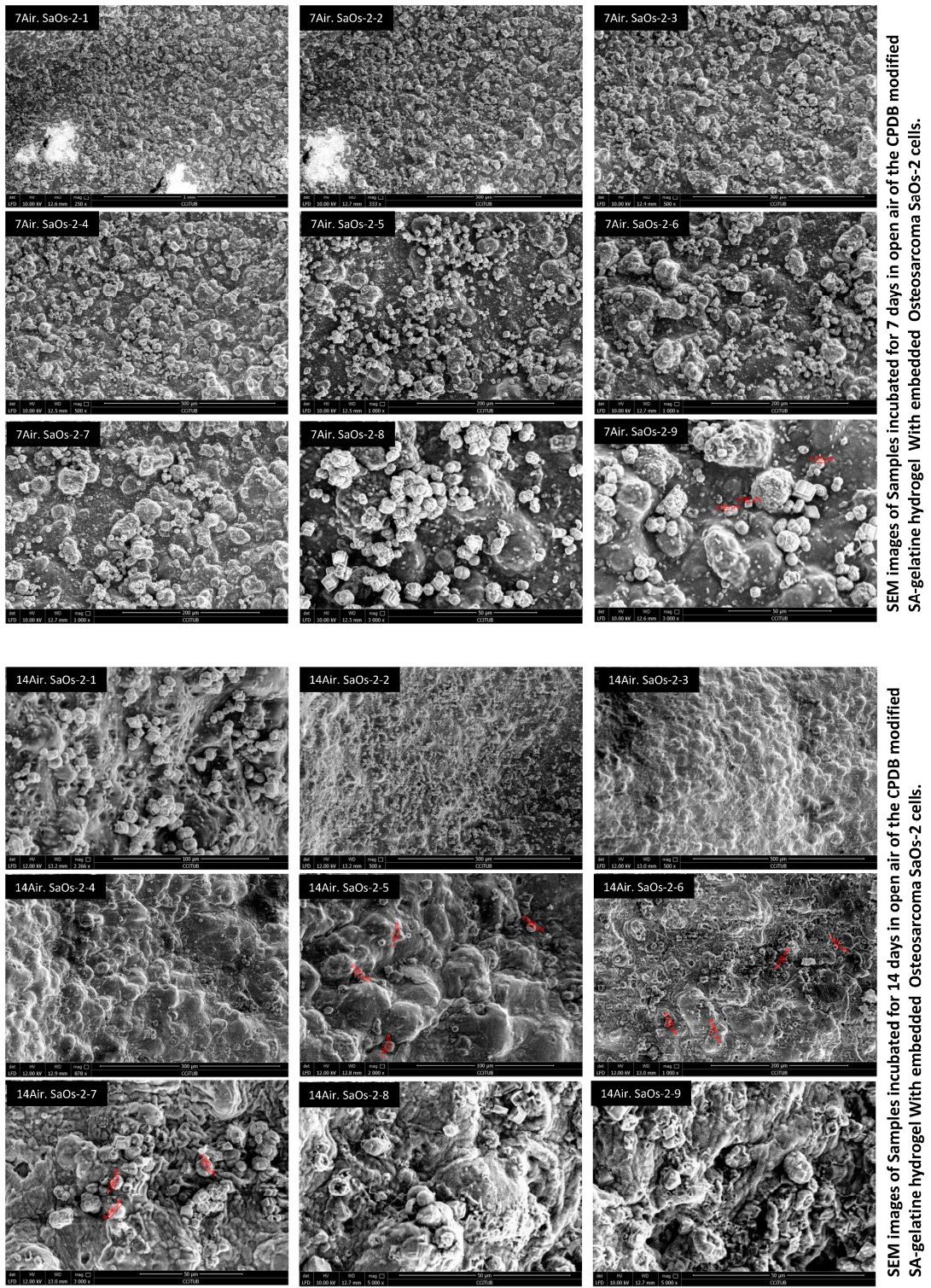

**Figure 7.** The SEM images of the bioink (optimized hydrogel with embedded osteosarcoma SaOs-2 cells) mineralization being incubated in open air in controlled but non-sterile conditions for 7 and 14 days, respectively. (7Air-SaOs-2-1 to 7Air-SaOs-2-9) The first group of the bioink samples incubated in air for 7 days at 37 °C, with varied scales ranging from 50 μm to 1 mm and with hydroxyapatite crustal size starting from 2 μm. (7Air-SaOs-2-1 to 7Air-SaOs-2-4) The dense distribution of the formed hydroxyapatite crystals across the hydrogel surface. (14Air-SaOs-2-1 to 14Air-SaOs-2-9) The bioink samples incubated in air for 14 days at 37 °C, with varied scales ranging from 50 μm to 500 μm, and with hydroxyapatite crystals size starting from 2 μm.

On the other hand, the hydrogel/bioink incubated in air (without and with cells, respectively) exhibited mild shrinkage in the dimensions of the 3D-printed samples after incubation in air for 14 days with a shrinkage of 2% from all directions from their size (the 3D-printed samples) post-cross-linking and pre-incubation in air for 14 days. However, another phenomenon was observed, which is the skewing or uneven rotation of the prints' upper surfaces, giving the shape of a hyperbolic paraboloid after complete mineralization, which will be exhibited and analyzed in detail in a future study.

## 4. Conclusions

In the current study, an optimized hydrogel was developed following a sustainable approach to minimize the materials and processes involved in developing self-mineralized hydrogel for multi-scale applications: as a bone replacement material and as a structural building material. The hydrogel developed mainly from three components of sodium alginate, gelatine, and calcium phosphate dibasic underwent a three-stage cross-linking process by cross-linking sodium alginate, and gelatine together; then cross-linking with calcium phosphate dibasic; and lastly, cross-linking the solution of calcium chloride and sodium carbonate. These thermally induced cross-linking reactions generated a viscose and stable hydrogel that has excellent injectability and shape retention as well. Three different concentrations of calcium phosphate dibasic showed higher rheological properties than the other tested concentrations. However, the 6% CPDB concentrations achieved the best rheological properties in terms of elasticity and rigidity. Although three different concentrations of epigallocatechin gallate were tested for enhancing the hydrogel elasticity and anti-microbial properties, the tested concentrations did not show any significance in these aspects. Thus, further investigations were conducted on the 6% CPDB-modified SA–gelatine hydrogel to detect its mineralization capacity in different environments and without and with cells, respectively. The SEM images proved the significant potency of the hydrogel to mineralize through forming hydroxyapatite platelets when incubated in the open air in non-sterile conditions for a duration of up to 14 days, with a congruent increase in mineralization over time. Similarly, the bioink with the embedded cells incubated in air in controlled but non-sterile conditions exhibited a similarly dense, mineralized material of hydroxyapatite crystals, which proves that the hydrogel is biocompatible and enables the cell activity of the embedded osteosarcoma SaOs-2 cells. Although the recorded mineralization in the case of the hydrogel incubated in FBS without and with cells recorded less mineralization density, the SEM images still exhibited hydroxyapatite crystals, showing the mineralization of the hydrogel when incubated in FBS. The degradability of the hydrogel was measured by the increase in its dimension when incubated in FBS for 14 days without and with cells. The results revealed a 5 to 20% increase in the dimensions of the hydrogel and bioink samples from all directions, especially in the case of bioink (with embedded cells). However, this ratio and time rate proves the slow degradation of the hydrogel, which enable its use in bone grafts and as a resistance structural building material. Furthermore, the hydrogel and bioink samples incubated in air did not show any swelling in dimensions but rather a shrinkage of 2% ratio from all directions after 14 days, with a detected skew in their surface morphology that will be evaluated and analyzed in a future study.

**Author Contributions:** Conceptualization, A.T.E. and Y.K.A.; Methodology, A.T.E. and Y.K.A.; Software, A.T.E. and Y.K.A.; Validation, A.T.E. and Y.K.A.; Formal analysis, A.T.E. and Y.K.A.; Investigation, A.T.E. and Y.K.A.; Resources, A.T.E. and Y.K.A.; Data curation, A.T.E. and Y.K.A.; Writing—original draft, A.T.E. and Y.K.A.; Writing—review & editing, A.T.E. and Y.K.A.; Visualization, A.T.E. and Y.K.A.; Supervision, A.T.E. and Y.K.A.; Project administration, A.T.E. and Y.K.A.; Funding acquisition, A.T.E. and Y.K.A. All authors have read and agreed to the published version of the manuscript.

**Funding:** This research received no external funding.

**Data Availability Statement:** All mandatory data are exhibited in the study, further raw data can be given upon request to the authors.

**Acknowledgments:** The authors would like to extend their sincerest thanks to Nuria Casals (head of the Pharmacology Unit in the Faculty of Medicine and Health Sciences at Universitat Internacional de Catalunya and the group leader of NeuroLipid Group at Universitat Internacional de Catalunya), and Samuel Bru Rullo for their kind and generous support in the logistic aspects of the current study. Equally, the authors would like to express their sincere thanks to the Bioengineering Institute of Technology at Universitat Internacional de Catalunya for their kind support in training and providing the used cell line; it would not have been possible to complete this research without their kind help. Last but not least, the authors extend sincere thanks to David Artiaga Torres and the CCiTUB at the University of Barcelona for their kind and professional support in the microscopy study in this work.

**Conflicts of Interest:** The authors declare no conflict of interest.

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
