# Peer review of "Biomimetic Approach for Enhanced Mechanical Properties and Stability of Self-Mineralized Calcium Phosphate Dibasic–Sodium Alginate–Gelatine Hydrogel as Bone Replacement and Structural Building Material"

_processes, doi:10.3390/pr12050944_

Round 1

Reviewer 1 Report

Comments and Suggestions for Authors

Herein, the authors conducted research into a new hydrogel material that could be used as a bone replacement material and structural building material. Its mechanical properties and antibacterial properties could be enhanced by modifying SA-gelatin hydrogel and adding PDB (coffee polyphenol derivative) to form hydroxyapatite crystals. The study compared hydrogels with different CPDB concentrations. The 6% CPDB-modified SA-gelatin hydrogel achieved the best rheological properties in terms of elasticity and hardness. Then, mineralization experiments were conducted on the optimal concentration of hydrogel, and its mineralization behavior in fetal bovine serum and air was observed to simulate the application environment in vivo and under non-sterile conditions, thus proving the effectiveness of the hydrogel. Fetal bovine serum and air can form hydroxyapatite crystals similar to bone tissue. Finally, the microstructure of the hydrogel under non-sterile conditions was observed by scanning electron microscopy, demonstrating its high antibacterial properties. This study therefore addresses the issue of developing hydrogel materials that can be used in bone replacement and structural construction by improving their properties and studying their behavior in different environments. It has important reference value for research in the fields of bone tissue engineering and biomaterials, and the experimental results have representative significance and certain application value, providing reference and inspiration for further research in related fields. The manuscript should be accepted after minor revision.

The comments are as follows:

1. There are few concentration choices in CPDB, and why only 5.5%, 6% and 6.5% are selected for comparison. The workload is slightly lacking.

2. The format needs to be modified, such as Table 2 should be revised the incorrect format. Also why so much underlines in page 20-21. All figures are not so clear, resolution (DPI) should be improved.  

Author Response

The Authors would like to express their sincerest thanks to the respectful reviewer for taking the time to sufficiently read and analyse the core objective and methods of the manuscript. the authors have read carefully and addressed the comments of the reviewer applying modifications where needed; as follows:

Comment Number 1: There are few concentration choices in CPDB, and why only 5.5%, 6% and 6.5% are selected for comparison. The workload is slightly lacking.

Answer Number 1: The experimental methodology tested 7 different concentrations of the Calcium Phosphate Dibasic added to the hydrogel base made of sodium alginate (4%) and gelatine (8%) as stated in the Materials and Methods Section ¨Rheological Properties of Alginate-Gelatine Hydrogel Enhanced with Different Concentrations of Calcium Phosphate.¨ between lines 422-425. (now highlighted in yellow in the text). This concentration ratio choice starting from 4% which is equivalent to the concentration of the used sodium alginate in the hydrogel base to balance the proposed three stage chemical crosslinking of the sodium alginate first by gelatine and second by the Calcium Phosphate dibasic, the increase in the concentration ratio of 0.5% of the Calcium Phosphate Dibasic was designed to enable the visibility of effects on the rheological and mechanical behaviour of the developed hydrogel. However, only the three concentrations of 5.5%, 6% and 6.5% passed the printability test (injectability test) with an adequate moderate viscosity that enabled their smooth 3D printing, Therefore these three concentrations were chosen to undergo the rheological tests. This Justification is provided in the Results and Discussion Section ¨3.1. Rheological Properties of SA-Gelatine Hydrogel Enhanced with Different Concentrations of Calcium Phosphate Dibasic¨ between lines 728 to 731 (now highlighted in yellow in the text). Thus, sense the other tested concentrations weren´t sufficient for printability, and congruent with the sustainable methods and processes employed in the current study, it wasn´t justified to include the other four different concentrations of the CPDB-modified hydrogel (4%, 4.5%, 5%, 7%) in further rheological or chemical testing. 

Comment Number 2: The format needs to be modified, such as Table 2 should be revised the incorrect format. Also why so much underlines in page 20-21. All figures are not so clear, resolution (DPI) should be improved.

Answer Number 2: The format was revised all over the manuscript; Table 2 headers were corrected to the correct format of units; the underlines in pages 20-21 were removed (these might have resulted from transferring the manuscript to the journal format, usually can happen some formatting errors), corrected and highlighted in Yellow in the text. The resolution of the images is a result of embedding the figures inside the word file, however, the original PPW high resolution figures will be provided separately with the corrected manuscript. 

Reviewer 2 Report

Comments and Suggestions for Authors

The authors try to develop a bioink for bone tissue engineering. But I did not find any biological characterization data can support their claim that the construct can be used as a bone replacement. I also can’t understand why the approach was named as a “Biomimetic Approach”. From the length of the paper, I believe the authors made a lot of efforts. However, this manuscript as original research had some basic problems that need to be addressed. For example, Abstract section was too long and had messy logic, it needed to be shortened and concise. In general, the schematic diagram (Figure 1) should show the novelty of your own research but cite other papers’ mechanism. Therefore, I suggested that the paper should be improved a lot before publishing.

Some detailed comments/suggestions:

1. Page 16, line 777-780, why is it suitable for bone application after drying 7 days in a non-sterile condition? Moreover, The storage modulus of the hydrogels was far from enough for bone tissue repair.

2. page 16, line 1210,  “different magnifications ranging from 50 to 200 μm” , 50 to 200 um was scale bar not magnifications.

3. Page 29, line 1298, 4 days or 14 days?

4. Page 30, line 1366, I am surprising that the cells had no contamination in open-air culture and can survive for 14 days. From the SEM images, I can't recognize which area was your materials and which part was your cells.

Comments on the Quality of English Language

need to be improved.

Author Response

The authors would like to thank the reviewer for providing these constructive comments. The authors have read them carefully and tried to apply modifications accordingly where applicable, and here are the answers for the reviewer comments:

Comment Number 1: The authors try to develop a bioink for bone tissue engineering. But I did not find any biological characterization data can support their claim that the construct can be used as a bone replacement.

Answer Number 1: The Authors would like to highlight that the study isn´t aimed to design a bioink for bone tissue engineering, but to design a simplified and optimized Hydrogel foe mineralized material that have multi-scale applications ranging from adopting it in bone tissue engineering materials due its capacity of mineralization as well as in architectural building materials applications. Thus, Indicating that the main focus in the current study is the hydrogel simplified and optimized composition to provide mineralization in various media and sufficient rheological properties in relevance to these proposed applications. Therefore, ¨biological characterization¨ in terms of studying the cells development and proliferation was not applicable within the scope of the current study; also because the embedding of living cells to develop a ¨bioink¨ isn´t compatible with the sustainable and simplified processes and materials employed in the current study methodology and objective since it requires specific conditions and devices of incubation on the long-run (in operation and performance). thus, the biological characterization that was adopted in the current study was minimized to scan the effect of the cells on the mineralization of the hydrogel which is the main aim (the hydrogel mineralization), to prove the possibility of adopting this hydrogel for further assessment in bone tissue engineering materials. Therefore, the Scanning Electron Microscopy was the best to serve this objective. Thus, this material-based research is more compatible with the multi-scale and multi-disciplinary applications proposed in the objective of the current study. However, Further detailed biological characterization focused on studying the behaviour of different strains of cells embedded in this hydrogel composition will be exhibited in a separate study due to the length and scope limitations of the current study

Comment Number 2: I also can’t understand why the approach was named as a “Biomimetic Approach”.

Answer Number 2: The Biomimetic approach refers here to the mineralization process of the developed hydrogel incubated in air or in FBS through the reaction between the outer shell (interface) of the hydrogel 3D prints with the media forming the hydroxyapatite crystals and while maintaining the elasticity of the hydrogel because of the effect of its inner part that isn´t totally mineralized. in other words because to refers to the cortical-cancellous hierarchical structural motifs of bone and the physical description of the mineralized bone tissue. This is explained in the manuscript in the result and discussion section in lines 1108 to 1116 (now highlighted in yellow in the text).

Comment Number 3: Abstract section was too long and had messy logic, it needed to be shortened and concise.

Answer Number 3: Done, The abstract section was revised and shortened where applicable. (highlighted modified parts in yellow in the text).

Comment Number 4: In general, the schematic diagram (Figure 1) should show the novelty of your own research but cite other papers’ mechanism.

Answer Number 4: The authors would like to highlight that Figure 1 diagram is not intended to show the novelty of the proposed chemical formula developed in the current study, but to exhibit the background from literature that supports the proposed approach in the current study of developing the hydrogel from the three main components: Sodium Alginate; Gelatine and Calcium Phosphate Dibasic in a three stage crosslinking process which was discussed thoroughly in the results and discussion section. while the diagram of Figure 1 is located in the introductory literature review section of the manuscript; thus, it can not be confused as a result of the current study but exhibit clearly the chemical basis of the developed approach in the current study from literature. 

Comment Number 5: Page 16, line 777-780, why is it suitable for bone application after drying 7 days in a non-sterile condition? Moreover, The storage modulus of the hydrogels was far from enough for bone tissue repair.

Answer Number 5: in the mentioned paragraph (now highlighted in yellow in the text): it is stated clearly that the hydrogel mineralizing in air in non sterile conditions within 7 days is adequate for its proposed multi-scale and multi-disciplinary applications; mainly as a structural architectural material in the built environment since it will not require special devices or controlled conditions to achieve its mineralization and anti-microbial traits. such an autonomously, cheaply and sustainably mineralized proposed hydrogel formula with anti-microbial traits qualifies for its integration in bone tissue engineering materials research as bone replacement material since under non-sterile conditions it exhibited high microbial-resistance as well as adequate mineralization and degradation rate, this make it suitable also for bone tissue engineering materials that one of its main concerns is in-situ in-vivo post-operative infection-control. This justification was added to the paragraph and highlighted in yellow in the text.  

concerning the reached rheological properties of the optimized hydrogel of the 6% CPDB-modified SA-gelatine hydrogel were discussed in the discussion section lines: 896-903 and compared with literature to prove that the reached stiffness of the hydrogel in the tested samples of 1mm thickness was congruent with the results obtained from other more complex alginate-gelatine based hydrogel compositions as GelMA; which proves that the developed hydrogel in the current study is more appealing for employing in bone tissue engineering materials being more sustainable in terms of less materials and processes consumption, while reaching the same results of other more complex hydrogels. furthermore, the storage modulus signifying elasticity was not the only parameter to judge the mechanical performance of the hydrogel, but the relation and equilibrium between elasticity and strength was the main criteria. furthermore, the storage modulus of the developed hydrogel in the current study was tested in absolute conditions meaning that for judging its approximation to the bone tissue material storage modulus; it needs to be tested in wet conditions as in-vivo conditions and in similar biomimetic structures as the hierarchical structural motifs of bone, which exceeds the scope and length of the current study and that will be exhibited in details in a future publication.  

Comment Number 6: page 16, line 1210,“different magnifications ranging from 50 to 200 μm” , 50 to 200 um was scale bar not magnifications.

Answer Number 6: Done, It was corrected the Figure5. caption on page 26 line 1210 (now highlighted in yellow in the text).

Comment Number 7: Page 29, line 1298, 4 days or 14 days?

Answer Number 7: The authors thank the reviewer for their comment. Done, it was corrected to 14 days and highlighted in yellow in the text. 

Comment Number 8: Page 30, line 1366, I am surprising that the cells had no contamination in open-air culture and can survive for 14 days. From the SEM images, I can't recognize which area was your materials and which part was your cells.

Answer Number 8: The authors would like to highlight that they did not mention that the cells were alive in open-air incubation after 14 days. the authors have highlighted that the main purpose was detecting the effect of the cells on the mineralization of the hydrogel as an inferred marker on the hydrogel cytocompatibility, however the cells viability, proliferation and differentiation or other biological markers on their behaviour are not within the objective of the current study that is focused on developing the material mainly without cells and with possibility of having cells, of course because without cells is even more sustainable approach for multi-scale applications. as explained above a further detailed study focused on the bioink cytocompatibility will be published in the future studying mainly the mineralization reaction kinetics under the bioactivity and metabolic pathways of the embedded cells. also, the authors would like to highlight that the embedded cells within the hydrogel were embedded with their adequate portion of medium within the hydrogel, and based on the biomimetic morphology of the mineralized hydrogel that forms a shell of mineralized hydroxyapatite platelets or spheres as exhibited in the SEM images, this shell contributed to the maintenance and preservation of the embedded cells for longer duration in the sense of creating a shield that surrounds the embedded cells while allowing them to benefit from the mineralization reaction results from the outer to the inner layers of the hydrogel. this justification was added to the mentioned paragraph highlighted in yellow. 

Concerning the cells position on the SEM images, they weren´t highlighted for two main reasons: 1) significant difference in scale between the cells scale of 100-200 nanometres and the examined mineralized material levels on the entire surface area of the hydrogel in microns; 2) the SEM was conducted to detect mineralization on the outer surface of the hydrogel which is the main interface for the mineralization reaction, which indicates less cell population on the surface being embedded within the body of the hydrogel. 

Reviewer 3 Report

Comments and Suggestions for Authors

This manuscript described the synthesis of hydrogel as bone replacement and structural building material and the characterization for mechanical properties and stability. This was well organized and characterized by the results of analytical tools. The authors should explain about what I pointed out below.

1. The authors did experiment on mineralization by incubation without and with cells for 7 and 14 days. Have the authors tried the experimental condition which is less than 7 days or more than 14 days? 

2. In this manuscript, the hydrogels were made by using physical bonding interaction such as coordination bond between metal cation and negative ion. Is it possible to use chemical bond for hydrogel in this system. 

3.  In conclusion, You mentioned that the 6% CPDB concentrations achieve the best rheological properties in terms of elasticity and rigidity. Please let me know the reason a little bit more details. 

Author Response

The authors would like to thank the reviewer for the constructive comments. The authors have read the comments carefully, answered them, and addressed them applying modifications in the manuscript where applicable.

This manuscript described the synthesis of hydrogel as bone replacement and structural building material and the characterization for mechanical properties and stability. This was well organized and characterized by the results of analytical tools. The authors should explain about what I pointed out below.

Comment Number 1: The authors did experiment on mineralization by incubation without and with cells for 7 and 14 days. Have the authors tried the experimental condition which is less than 7 days or more than 14 days? 

Answer Number 1: in the current study the authors have designed the experimental study based on recommendations and methods of previous literature; therefore, the mineralization detection was performed after 7, and 14 days of incubation in open-air and in FBS respectively for the hydrogel and the bioink respectively. This was justified throughout the manuscript as exhibited in the introductory literature review section in lines 317 to 320 (now highlighted in yellow in the text), showing that 7 days period was recommended by Lee, et al., (2011) showing that 3-7 days of incubating calcium phosphate modified alginate hydrogels increases their compressive strength. As well in the discussion section showing the increased density of the formed hydroxyapatite crystals in the 7-days interval (after the first 7 days of incubation and in the following 7-days duration until the 14th day; more pronounced in the hydrogel and the bioink the incubation in air.

Comment Number 2: In this manuscript, the hydrogels were made by using physical bonding interaction such as coordination bond between metal cation and negative ion. Is it possible to use chemical bond for hydrogel in this system. 

Answer Number 2: in the current study the authors have proposed a three-stage thermally induced complex chemical crosslinking to the proposed hydrogel. This sequence of the crosslinking was designed to crosslink the Sodium Alginate with Gelatine, then Crosslinking the SA-gelatine mixture with the calcium phosphate dibasic, finally crosslinking the hydrogel with the Calcium Chloride+ Sodium Carbonate solution. Within this physical-chemical reaction sequence, chemical bonding between the various reactive groups of the Sodium Alginate, Gelatine and Calcium Phosphate were irreversibly formed which proves that the chemical bonding happening between the three-component hydrogel catalyzed by temperature and cation and ion donner agents. Furthermore, the incubation of the optimized hydrogel in air or in FBS have played a final mineralization reaction to form the hydroxyapatite crystals as proved in the exhibited SEM results which prove as well the non-reversable chemical crosslinking happened to the hydrogel. Which is going beyond literature that use simple either physical or chemical crosslinking by using both in a complex successive interactions as explained thoroughly in the discussion section in pages 24, and 25. As well as explaining its literature background in the introduction section pages 2 to 7, explaining in details the chemical bonding reactions that crosslink Sodium Alginate, Gelatine and calcium phosphate which were the basis that the authors build upon their proposal of the three-components, three stage physical-chemical crosslinking.

Comment Number 3: In conclusion, You mentioned that the 6% CPDB concentrations achieve the best rheological properties in terms of elasticity and rigidity. Please let me know the reason a little bit more details. 

Answer Number 3: The authors built their conclusion based on the rheological properties results exhibited in the results and discussion section ¨3.1. Rheological Properties of SA-Gelatine Hydrogel Enhanced with Different Concentrations of Calcium Phosphate Dibasic¨ page 16 as exhibited in Figure 2 and page 17 as exhibited in Table 1showing the comparison between the different tested concentrations of the calcium phosphate dibasic modified hydrogels in terms of rheological properties of storage and loss modulus as well as in the ratio between these values. Thus, as explained in the manuscript in these sections, it was proven that the 6% CPDB modified hydrogel achieved the best equilibrium between elasticity and stiffness which was the main criteria for a self-mineralized material that can be applied in bone tissue engineering as well as in architecture as a building material.

Round 2

Reviewer 2 Report

Comments and Suggestions for Authors

The authors have addressed my concerns.

Comments on the Quality of English Language

Please double check the whole manuscript to make sure there is no spelling mistake.

Author Response

The authors would like to thank the reviewer for their constructive comments.

The manuscript was revised for English language, style and grammar checking, and modifications were applied where needed. 

The authors extend their sincere thank to the reviewer.